# Bio-inspired magnetic-driven folded diaphragm for biomimetic robot

Dezhao Lin [1], Fan Yang [1]✉, Di Gong [1,2] & Ruihong Li [1]

Functional soft materials, exhibiting multiple types of deformation, have shown their potential/abilities to achieve complicated biomimetic behaviors (soft robots). Inspired by the locomotion of earthworm, which is conducted through the contraction and stretching between body segments, this study proposes a type of one-piece-mold folded diaphragm, consisting of the structure of body segments with radial magnetization property, to achieve large 3D and bi-directional deformation with inside-volume change capability subjected to the low homogeneous magnetically driving field (40 mT). Moreover, the appearance based on the proposed magnetic-driven folded diaphragm is able to be easily customized to desired ones and then implanted into different untethered soft robotic systems as soft drivers. To verify the above points, we design the diaphragm pump providing unique properties of lightweight, powerful output and rapid response, and the soft robot including the bio-earthworm crawling robot and swimming robot inspired by squid to exhibit the flexible and rapid locomotion excited by single homogeneous magnetic fields.

Ranging from multiscale and functional self-assemblies to morphology control, complex locomotion strategies, and autonomous feedback dynamics, nature has provided us with numerous fascinating examples for biomimetic research[1–4]. Recently, great efforts have been made in mimicking organic species to devise soft robotic systems with novel functionalities and accessibility to increasingly challenging spaces[5–11]. A variety of stimuli, including electric fields[12,13], chemicals[14], heat[15], light[16], pressure[17], or magnetic fields[18], have been successfully used to control morphology changes for locomotion or object manipulation using soft active materials. Among them, magnetic fields, which can easily and harmlessly penetrate most biological and synthetic materials and offer a safe and remoting actuation method, have great application potential for remote soft actuators and robots[19]. Recently, the designs of magnetically soft actuators and robots inspired by different biomimetic behaviors, such as snake[20], inchworm[21], bird[22], jellyfish[23], spermatozoid[24], cilium[25], beetle[26], tadpole[27], brittle star[28], and turtle[29], continue to appear and become one of the research hotspots in the area of soft robots.

In order to achieve complex locomotion and deformation required by the magnetically biomimetic soft robot, different methods of manufacturing magnetic-driven soft materials, including splicing of pieces with different magnetic properties and additive manufacturing technology (4D printed), have been reported[30–32]. Apart from the advanced magnetic property designs, the advanced magnetic stimuli, such as the rotational magnetic field and robot-controlled electromagnet, have been made effort to conduct the locomotion of magnetically soft robots[33–35]. Meanwhile, many kinds of magnetic-driven soft actuators have been reported, including the origami structure[22,30,36–38], but most of them can only achieve the changes in shapes[30], angles[36], and/or lengths[38], and cannot realize the relatively large inside-volume changes with high strength, which may limit their potential practical application as soft actuators (drivers), such as the micro-pump. Besides, to mimic the deformation of some tissue species, such as the heart and muscle, their surface should be expanded or compressed during the deformation process[39,40]. It leads to a large elastic resistance, and then restricts the deformation range or needs a

[1]Research Center for Intelligent Materials and Structures (CIMS), College of Mechanical Engineering and Automation, Huaqiao University, Xiamen, Fujian, P.R. China. [2]Institute of Extremely-Weak-Magnetic-Field Massive Scientific Instrumentation Facility, Hangzhou, Zhejiang, P.R. China. ✉e-mail: xmyf@hotmail.com

powerful magnetic field, which may limit the application of the magnetically soft robot.

Therefore, in this study, a kind of magnetic-driven folded diaphragm, with a one-piece molded and simple manufacturing procedure, and large, 3-D, and bi-direction deformation subjected to simple homogeneous and related low-strength magnetic fields, is proposed. This kind of folded diaphragm with different radial magnetization properties is able to realize large inside-volume changes with high strength, which is inspired by the locomotion of earthworms generated by the contraction and stretching between body segments. Based on the extensive experimental tests, the proposed folded diaphragm shows some distinctive features. Firstly, the folded diaphragm can achieve large deformation under a 40 mT magnetic field, which is mainly because of the deformation with low elastic-resistance force due to the folded arrangement. Secondly, the folded diaphragm can generate 3D and bi-directional deformations due to the different radial magnetization properties in each folded segment. Finally, the folded diaphragm can be fabricated using the simple one-piece molded method, easily customized to different shapes based on the practical requirements, and then implanted into different untethered soft robotic systems as soft drivers (actuators). To verify the advantage of the proposed magnetically folded diaphragm, two different types of typical applications (the diaphragm pumps and soft biomimetic robots) excited by the single homogeneous magnetic field have been designed. The diaphragm pump system can provide powerful output and rapid response with a lightweight property. The soft biomimetic robots, which include the bio-earthworm crawling robot and the swimming robot inspired by squid, exhibit flexible and rapid

locomotion. Considering the outstanding advantages of the proposed folded diaphragm mentioned above together with the essence of the wireless magnetic control, it shows great application potential for soft actuators and biomimetic robots.

## Results

### Operational principle of the magnetic-driven folded diaphragm

The locomotion of Annelida, such as earthworms, is achieved through the contraction and stretching between body segments, as shown in Fig. 1a. Inspired by earthworm's motion pattern, a folded diaphragm consisted of a couple of segments, which is a kind of composite of the hard-magnetic particle and silicone rubber. The composite materials are poured into the shape fabrication mold, as shown in Fig. 1b until it is fully curing. After that, the shaped composites are placed in the magnetization mold shown in Fig. 1c, to be magnetized to obtain different angles of magnetization direction in the radial direction on different segments, which include two types (Type+ and Type−) of diaphragms with opposite magnetization direction, as shown in Fig. 1d, e.

The segment deformation of the proposed folded diaphragm can be assumed as a rotation joint driven by the magnetic torque and resisted by the elastic torque applied on the joint, as shown in Fig. 1d, e. Let us utilize the Type+ diaphragm as an example to illustrate the basic working mode of the proposed folded diaphragm. Excited by the axially upward magnetic field, the interaction torque between the magnetization of segments and the external magnetic field generates the counterclockwise torque and attempts to align the segment to parallel with the direction of the external magnetic field, and then the

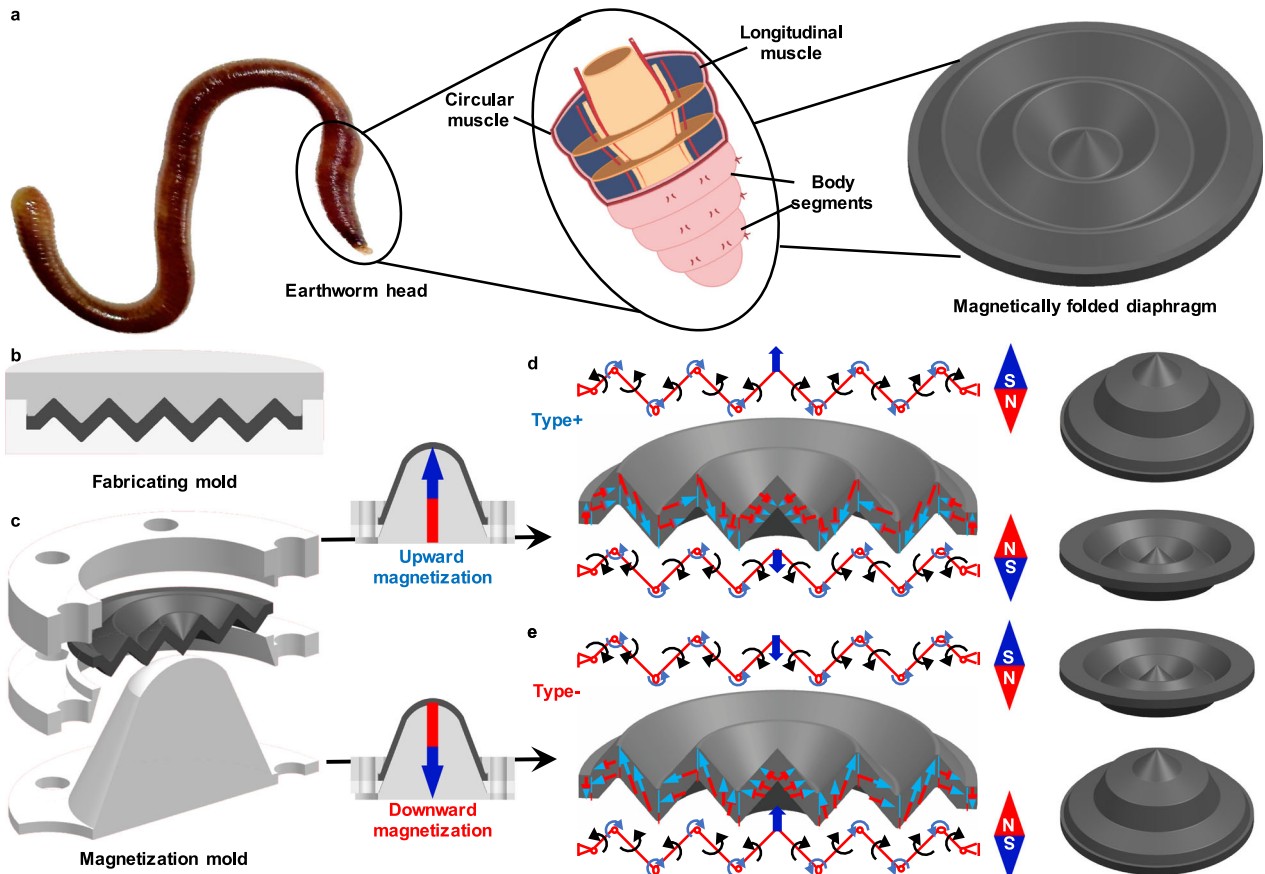

**Fig. 1 | The folded diaphragm inspired by the segment structure of earthworm.** **a** Bio-inspired magnetic-driven folded diaphragm from an earthworm. **b** Fabrication mold. **c** Magnetization mold of the magnetic-driven folded diaphragm. **d** Magnetization characteristics and operational principle of the magnetic-driven folded diaphragm with upward magnetization direction (Type+ diaphragm). **e** Magnetization characteristics and operational principle of the magnetic-driven folded diaphragm with downward magnetization direction (Type− diaphragm).

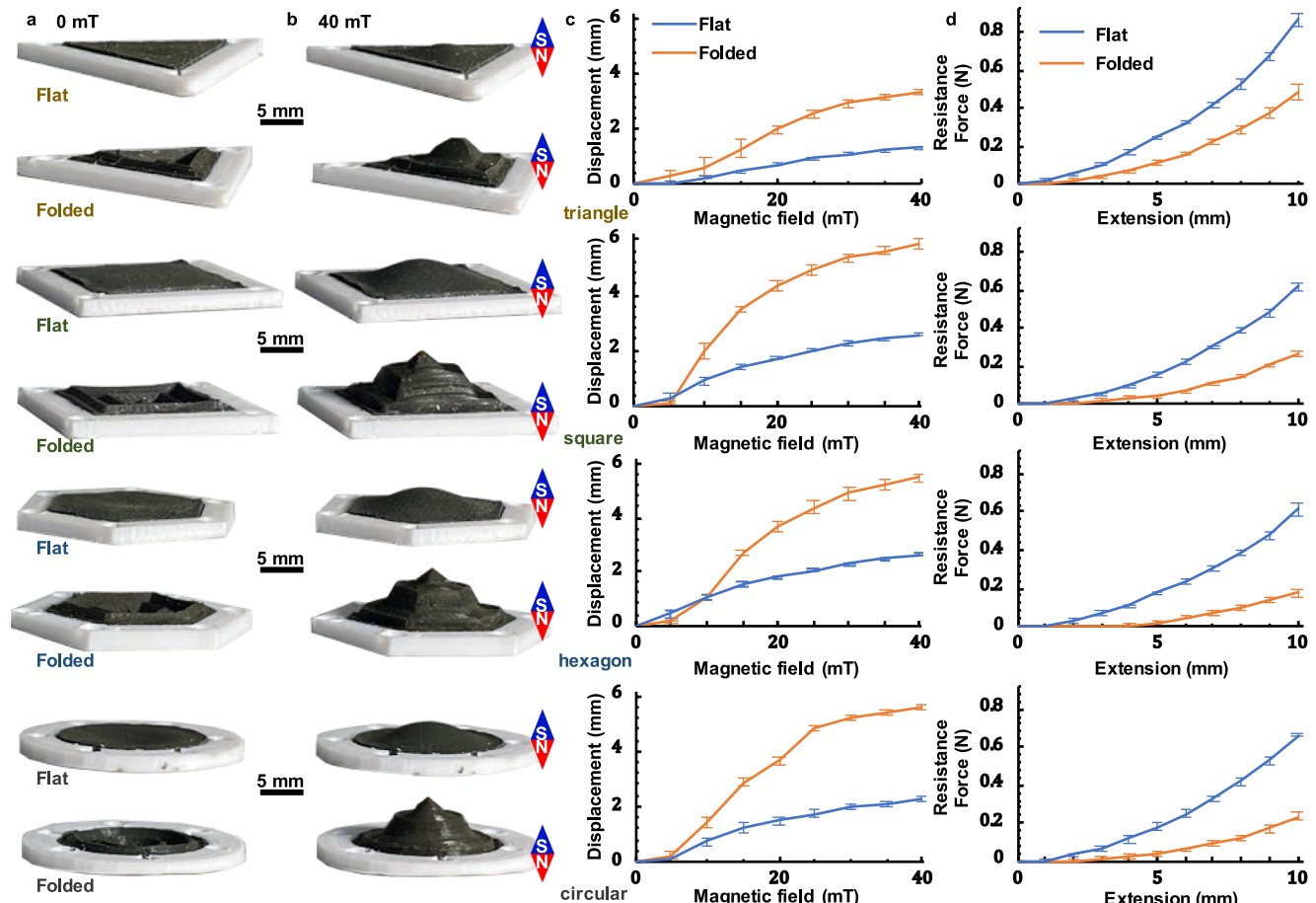

**Fig. 2 | Mechanical properties of the folded diaphragm. a** Customizable shape of the magnetic diaphragm in triangle, square, hexagon, and circular. **b** comparisons of deformation range between the folded and flat magnetic diaphragm under the 40 mT magnetic field by vision. **c** comparisons of the displacement (average value and standard deviation of 9 cycles measurement) in the top point between the folded and flat magnetic diaphragm under different magnetic fields (range from 0–40 with 5 mT increments). **d** comparisons of the elastic-resistance force of the diaphragm center between the folded and flat magnetic diaphragm under different displacements ranging from 0–10 mm with 1 mm increment (average value and standard deviation of 9 cycles measurement).

segment achieves counterclockwise alignment rotation, which generates upward deformation from the point of the diaphragm with the combination of alignment of each segment. Inversely, under the axially downward magnetic field, the clockwise magnetic torque generates the downward deformation, as presented in Fig. 1d. The whole deformation direction of the Type− diaphragm is opposite to those of Type+ one, as shown in Fig. 1e. Therefore, under the outer contour constraint condition, the magnetic-driven folded diaphragm can easily achieve 3D deformation driven by the homogenous magnetic field, which is caused by the magnetic torque for each segment around the rotation joint. The detailed operational mechanism is presented in Supplementary Note 1 and Supplementary Fig. 1. Here, it should be noted that due to the folded arrangement, the surface area of each segment has not been significantly changed during the deformation process, and then only a small elastic-resistance force caused by the elastomer extension is generated during the deformation procedure, which provides the possibility to generate large deformation with the small exciting magnetic field.

## Mechanical properties

The deformation characteristics are one of the most important indexes to evaluate the performance of the proposed folded diaphragm. Furthermore, as mentioned before, the folded diaphragm can be customized to different shapes to meet the requirement of practical applications. Therefore, in this section the deformation properties of triangle, square, hexagon, and circular shapes of the proposed folded diaphragm, as shown in Fig. 2a, are evaluated based on the designed experimental, and compared with the normal magnetic flat diaphragm. In this study, in order to verify the reproducibility of the large deformation capability subjected to the low magnetically driving field, three samples have been fabricated for each type of diaphragm shown in Fig. 2a, and each sample has been tested for three cycles. Therefore, each type of diaphragm shown in Fig. 2a has been tested for 9 cycles total. Figure 2b clearly illustrates the outstanding deformation properties of the proposed magnetic folded diaphragm compared with those of the flat one, which should overcome the large elastic-resistance force due to the extension of the flat diaphragm, under the same magnetic field. For example, the deformation of a circular folded diaphragm (5.568 ± 0.118 mm) is 146% larger than that of a circular flat diaphragm (2.267 ± 0.114 mm) under the same magnetic field, as shown in Fig. 2c, which is mainly because of the much small elastic-resistance force generated from the folded diaphragm compared with that of the flat one under the same deformation condition, as shown in Fig. 2d. The large deformation property of the folded diaphragm can be observed from the simulation result, too, as shown in Supplementary Fig. 2a, b in the Supplementary Note 2. Here, it should be noted that the results illustrated in Fig. 2c, d include the average value and standard deviation of 9 testing cycles for each type of diaphragm, and the maximum standard deviation, which is around 1.397 ± 0.323 mm, is

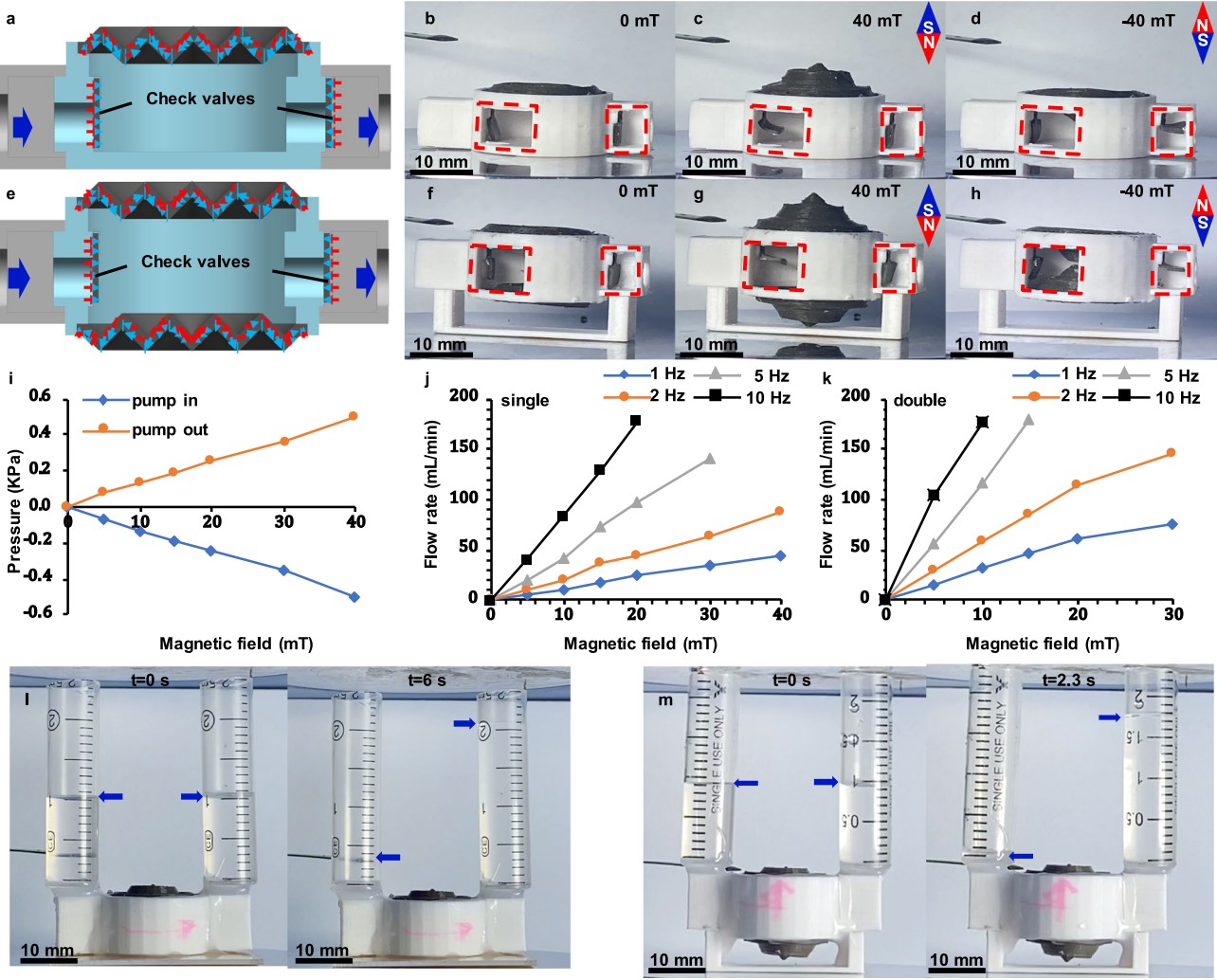

**Fig. 3 | Diaphragm pump based on the folded diaphragm. a** Schematic diagram of the single diaphragm pump. **b–d** The pictures of a single diaphragm under 0, 40, and −40 mT magnetic fields, respectively. **e** Schematic diagram of the double diaphragm pump. **f–h** The pictures of double diaphragms under 0, 40, and −40 mT magnetic fields, respectively. **i** The pump-in and pump-out pressure properties of the single diaphragm pump. **j** and **k** The ideal pump-in and pump-out flow rate properties of single diaphragm pump and double diaphragm pump, respectively. **l** Unidirectional fluid transfer function of the single diaphragm pump under −40 to 40 mT harmonic magnetic field with 1 Hz frequency. **m** Unidirectional fluid transfer function of the single diaphragm pump under −20 to 20 mT harmonic magnetic field with 5 Hz frequency.

located in the deformation curve for the triangle folded diaphragm subjected to 15 mT magnetic fields.

Supplementary Note 3 summarizes the deformation characteristics of the magnetic-driven folded diaphragm subjected to multiple conditions, which include the diaphragm thickness, folded degree, number of segments, hardness of the silicone elastomer, the weight ratio of NdFeB particles to silicone elastomer and the magnetization degree. The fabrication conditions are presented in Supplementary Table 1 and the comparison results of different fabrication conditions are shown in Supplementary Figs. 3–8.

### Implementation in diaphragm pump

The large deformation property of the proposed folded diaphragm can be utilized to design the diaphragm pump. In this study, the single diaphragm pump consisting of a single proposed Type+ folded diaphragm as presented in Fig. 1d, is developed, as shown in Fig. 3a–d. The single diaphragm pump consists of two opposite magnetic sheets as check valves to cover/open the pump inlet/outlet ports, and one Type+ folded diaphragm glued to the upper surface of the PLA pump body to suck/push the hydraulic medium. The pump magnetization

characteristics have been shown in Fig. 3a. The double diaphragm pump adds one Type− diaphragm glued to the bottom surface of the PLA pump body, as shown in Fig. 3e–h. The detailed dimensions of both pumps are provided in Supplementary Fig. 9, and their weights are shown in Supplementary Fig. 10. Under vertical upward magnetic field conditions, the magnetic sheet in the inlet (outlet) port bends rightward (leftward) to open (close) the inlet (outlet) port and vice versa. At the same time, the deformation of the folded diaphragm (s) expands the cavity volume, and then sucks the flow (gas/water) into the pump cavity, as shown in Fig. 3c, g. Under vertical downward magnetic field conditions, the deformation of the folded diaphragm compresses the cavity volume, and then pumps the flow (gas/water) out from the pump cavity, as shown in Fig. 3d, h. The deformation simulation result of the simplified model of the single diaphragm pump is shown in Supplementary Fig. 2c. Compared with the conventional magnetic-driven pump, which should be driven by the gradient and strong magnetic field[41,42], the proposed diaphragm pump designed based on the magnetic-driven folded diaphragm can realize the large inside-volume change with appreciable loading capability stimulated by the homogeneous and low magnetically driving field,

which makes it have considerably protentional application in the area of the diaphragm pump. Besides, compared with the conventional soft actuator-based pumps designed based on dielectric elastomers[43,44], the proposed magnetic-driven diaphragm pump can effectively avoid the possible electrical connectors for application in a hydraulic environment.

Ignoring the leakage of the check valve, the output pressure, and flow rate can be modified by the amplitude of the applied magnetic field, and the flow rate is also related to the frequency of the applied magnetic field, as shown in Fig. 3i–k (the average value and standard deviation data are shown in Fig. 3i–k, and summarized in the Supplementary Tables 2–4). It can be found that under the −10 to 10 mT harmonic magnetic field with 10 Hz frequency, the double diaphragm pump can provide $178.1 \pm 3.5$ mL min$^{-1}$ flow rate, which provides rapid response property. The diaphragm pump merges the lightweight with powerful output properties and then has large specific pressures ($-167.3 \pm 6.7$ kPa kg$^{-1}$, the ratio of maximum pressure to the weight of single diaphragm pump (3.01 g)) under the 40 mT magnetic field and specific flow rates ($-61394 \pm 748$ mL min$^{-1}$ kg$^{-1}$, the ratio of maximum flow rate to the weight of double diaphragm pump (2.94 g)) under −15 to 15 mT harmonic magnetic field with 5 Hz frequency superior to those reported magnetic micro-pumps[45]. Besides, the performance of the proposed diaphragm pump is related to the deformation properties of the proposed folded diaphragm directly, as shown in Supplementary Note 3. Furthermore, the cooperation between the magnetic diaphragm and sheets (as a check valve) makes the proposed diaphragm pump can be excited by single stimuli without any additional valve device. Figure 3l, m shows the desirable functions for unidirectional fluid transfer under different excitation magnetic fields, and the pumping process is shown in Supplementary Movie 1.

### Bionic crawling robots

Inspired by the earthworm, a crawling robot is designed based on the proposed magnetic folded diaphragm, as shown in Fig. 4. The main structure of the proposed crawling robot consists of the following parts: (1) The outer edge of the proposed magnetic folded diaphragms (Type+ and Type−) glued on the PLA basics (A and B), and then connected by a PLA rod on the center of both diaphragms, as shown in Fig. 4a–c, to achieve the contraction and stretching functions of the earthworm's muscle; (2) 4 wheels installed in each PLA basics to provide guiding purpose; (3) 8 identical magnetic sheets with thickness magnetization (guiding sheets) installed in each PLA base to provide the function of earthworm bristles. Here, it should be noted that the series structure can be utilized to achieve the large extension range, as shown in Fig. 4b. The cooperation of the proposed diaphragms and the guiding sheet can realize the crawling function, as illustrated in Supplementary Movie 2.

The main working procedures are the following: (1) without applied external magnetic fields, all magnetic parts (diaphragms and sheets) maintain their initial condition, as shown in Fig. 4d, g; (2) Applied magnetic fields shown in Fig. 4e, h, the sheets in the tail part become stiff and then restrict the movement of the tail part. At the same time, the sheets in the head part become curled and then allow the movement of the head part; (3) the diaphragms are extended; (4) the cooperation of the sheets and the diaphragms allow the head part to move forward; (5) Applied opposite magnetic fields, the movements of sheets and diaphragms are also opposite, and then shrink the whole structures, as illustrated in Fig. 4f, i; (6) the whole procedure can be repeated under applied symmetric periodic magnetic fields, as presented in Supplementary Movie 2. The deformation simulation result of the simplified model of the crawling robot is shown in Supplementary Fig. 2d. Besides, due to the soft connecting characteristics between each segment, the proposed crawling robot can achieve much more flexible motion functions, such as climbs, turns, and over the triangular slope, which is superior to other magnetic crawling

robots acted by the rotational magnetic field[33], as shown in Supplementary Movie 2. Meanwhile, the locomotion speed of the crawling robot can be modified through the intensity and frequency of the applied magnetic field, as shown in Supplementary Movie 3.

As shown in Supplementary Movie 4, under the −40 to 40 mT harmonic magnetic field with 2 Hz frequency derived by the solenoid magnetic field generator, it takes about 5 s to crawl through the 150 mm channel (about 30 mm s$^{-1}$ speed), which is more rapid than the other report magnetically crawling robot[46]. Furthermore, due to the serially stackable characteristic of the proposed crawling robot (shown in Fig. 4b), under the same magnetic excitation, the more stacks, the larger contraction and stretching ranges, and then the more rapid movement, as shown in Fig. 4h, i.

### Bionic squid swimming robots

As mentioned before, the proposed diaphragms, as a driving part, can be embedded and utilized to design different kinds of systems. In this section, it is designed to mimic the squid's water jet propulsion. As we all know that the squid's tentacles unfold and fold procedures are related to the expansion and contract of the body cavity, respectively, and then complete the water suck and jet procedures, as shown in Fig. 5a, b. Inspired by the squid's movement, a swimming robot based on the proposed magnetic-driven folded diaphragm is designed, as shown in Fig. 5c. The proposed swimming robot consists of one folded diaphragm to achieve the expansion and contract of the body cavity, 10 pieces of magnetic sheets with thickness direction magnetization mimicking the squid's tentacles to provide the restrict force in the sucking water procedure, five pieces of magnetic sheets with lengthwise magnetization mimicking the check valve function of squid's funnel to co-operate with the diaphragms, one jet pipe, and one shell. The magnetization characteristics of the above components are shown in Fig. 5d.

Appling the external magnetic field, as shown in Fig. 5e, g, the 10 pieces of magnetic sheets with thickness direction magnetization become stiff to restrict the movement, and the diaphragm extents with check valves in the open condition to allow the sucking water procedure. The deformation simulation result based on the simplified model of the swimming robot is shown in Supplementary Fig. 2e. The whole procedure can be reversal under the applied opposite magnetic field, as shown in Fig. 5f, h to complete the jet procedure. The sucking and jetting procedures can be repeated to complete the movement of the swimming robot under the applied period excitation signal with a strong correlation to the excitation frequency. Figure 5e, f illustrates the sucking of the jet procedures of the pumping structure in the designed swimming robot, and detailed information is provided in Supplementary Movie 5, in which the −40 to 40 mT harmonic magnetic field with 2 Hz frequency derived by the parallel magnetic field generator. The bionic swimming robot can achieve snorkeling, diving, and horizontal diving, as provided in Supplementary Movie 6, from which it can be found that it takes about 6.2 s to cross the 150 mm channel (24 mm s$^{-1}$ speed) under the −40 to 40 mT harmonic magnetic field with 2 Hz frequency derived by the solenoid magnetic field generator.

### Discussion

In this study, we propose a type of one-piece-mold folded diaphragm with radial magnetization property, which can achieve large 3D and bi-directional deformation with large inside-volume change capability subjected to the low homogeneous magnetically driving field with the easily fabricating method of one-piece mold. Based on the proposed diaphragm, we design three types of applications: (i) For the diaphragm pump application, under the small magnetic field excitation, the diaphragm pump merges the unique properties of lightweight and powerful output, enabling them to have larger specific pressures ($167.3 \pm 6.7$ kPa kg$^{-1}$) and specific flow rates ($-61394 \pm 748$ ml min$^{-1}$ kg$^{-1}$) superior to those magnetic micro-pumps together with the desirable functions for unidirectional fluid transfer; (ii) the crawling robot

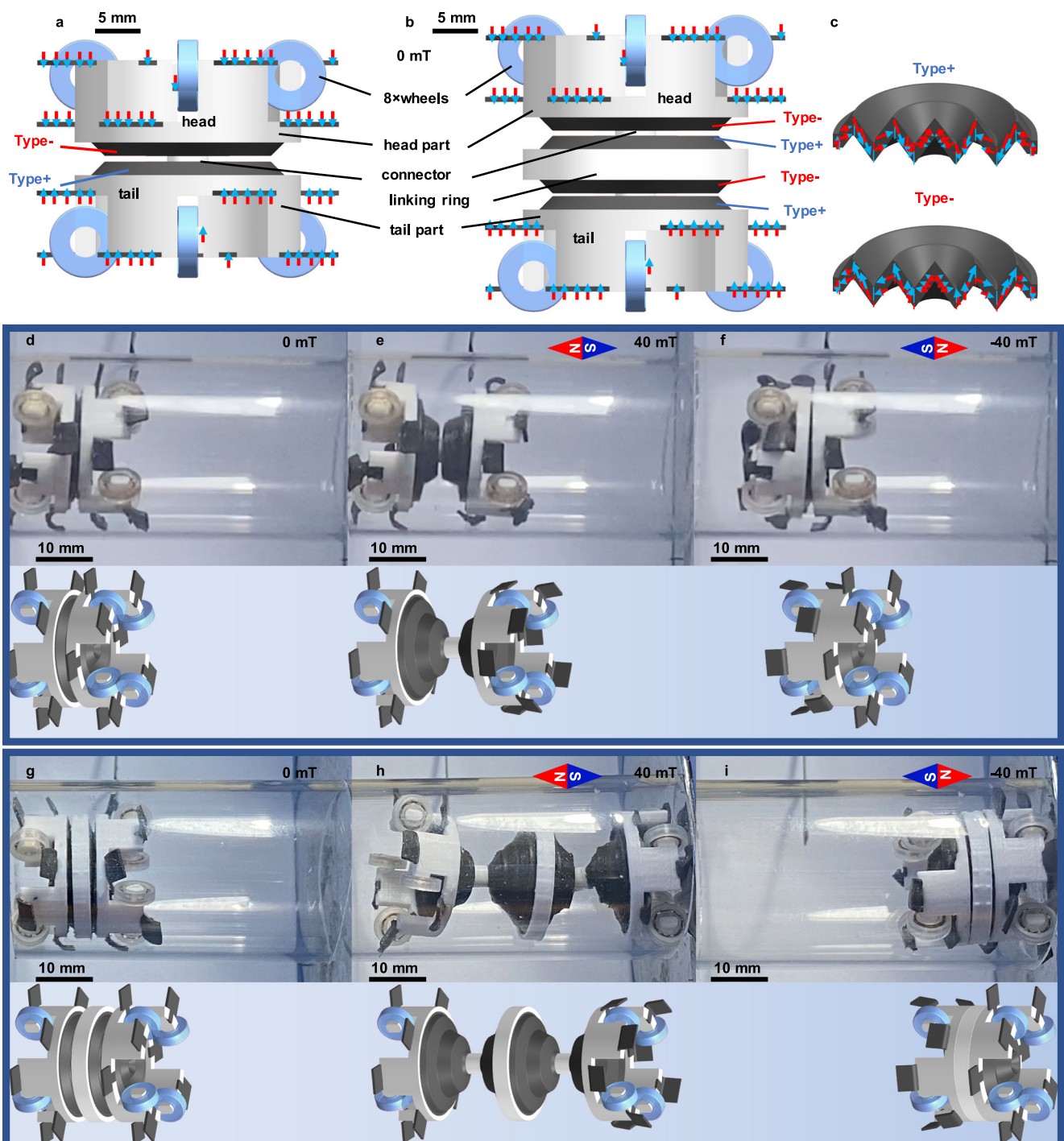

**Fig. 4 | Crawling robot inspired by the earthworm. a** Schematic diagram of the single-section crawling robot. **b** Schematic diagram of the double-section crawling robot. **c** Components' magnetization characteristics. **d**–**f** Movement patterns of single-section crawling robot under 0, 40, and −40 mT magnetic fields, respectively. **g**–**i** Movement patterns of double-section crawling robot under 0, 40, and −40 mT magnetic fields, respectively.

mimicking the earthworm can achieve climbs, turns and over triangular slop locomotion under single homogenous magnetic field, and crawl through 150 mm channel with 30 mm s⁻¹ average speed, which also can be improved by the stackable body segments. (iii) inspired by the locomotion of water jet propulsion of squid, the swimming robot can achieve snorkeling, diving, and horizontal diving with 24 mm s⁻¹ speed −40 to 40 mT under the harmonic magnetic field with 2 Hz frequency. Together, considering the magnetically wireless control and single homogeneous driving magnetic field, it shows great application potential of the soft actuator and biomimetic robot.

## Methods
### Material fabrication
The magnetic diaphragm proposed in this research consisted of neodymium–iron–boron (NdFeB) magnetic microparticles (MQP-15-7, Magnequench; average diameter: 5 μm, density: 7.61 g cm⁻³) and silicone elastomer matrix (Ecoflex 00–20) with a mass ratio of 1:1 (NdFeB particles to Ecoflex 00–20). After stirring for 3 min by a stirrer, the NdFeB and elastomer mixture materials were poured into the bottom side of the mold and then placed in a vacuum cylinder for 4 min in order to eliminate bubbles. Finally, after covering the top

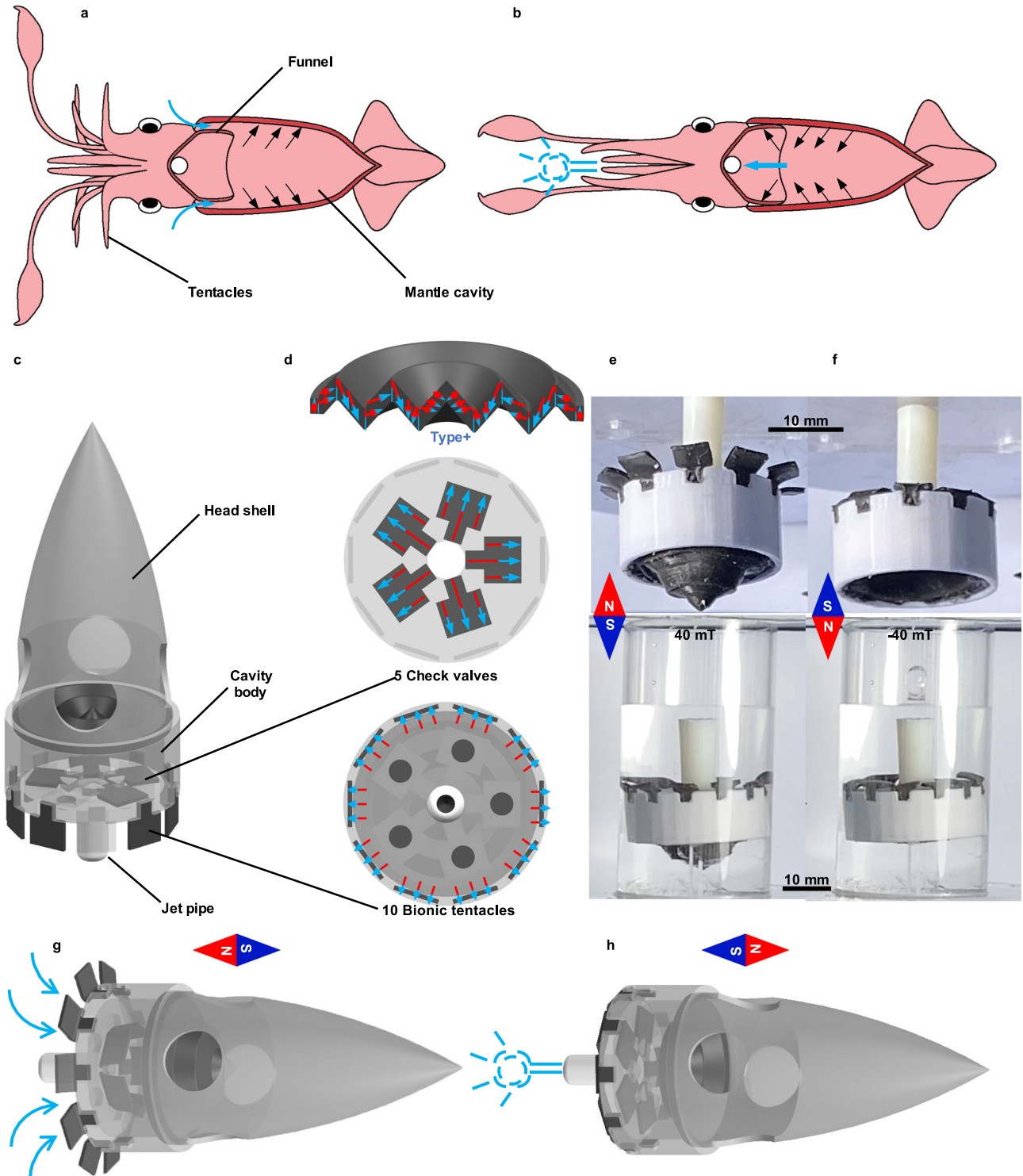

**Fig. 5 | Swimming robot inspired by squid. a** The water-filling process of squid's mantle cavity. **b** The squid's water jet propulsion. **c** Schematic diagram of a bionic squid swimming robot. **d** Magnetization characteristics of the magnetic folded diaphragm, bionic tentacles, and check valves. **e** Picture of cavity water filling water of bionic squid swimming robot (40 mT). **f** Picture of water jet of bionic squid swimming robot (−40 mT). **g** Sucking and **h** Water jetting schematic diagram of swimming process of swimming robot under the harmonic magnetic field.

side of the mold, the mixture materials were kept in the mold for 180 min until fully cured, and then the diaphragm with folded structure was fabricated. The fabricating process was shown in Supplementary Fig. 11a. Due to different requirements of the actuator shape, the diaphragms (folded and flat) with different shapes (triangle, square, hexagon, and circular) were fabricated (as shown in Supplementary Fig. 11a), which were used to verify the outstanding

deformation property of the magnetically folded diaphragm compared with the flat diaphragm (their dimensions are shown in Supplementary Fig. 12). In this study, 14 types of prototypes (three samples for each prototype) with different diaphragm thickness, folded degree, number of segments, hardness of the silicone elastomer, the weight ratio of NdFeB particles to silicone elastomer and the magnetization degree, as summarized in Supplementary Table 1,

were fabricated and tested to investigate the deformation and elastic-resistance force properties of the proposed folded diaphragms. Here, it should be noted that each sample (3 samples for each prototype) was tested 3 times, and then total of 9 tests for each type of prototype, as summarized in Fig. 2 and Supplementary Figs. 3–8, to verify the reproducibility of the deformation properties.

### Diaphragm magnetization

The magnetization process aimed to generate radial magnetization direction around the diaphragm center. The magnetization molds for four types of shapes of the diaphragm were shown in Supplementary Fig. 11b. The first step was to fix the outer edge of the diaphragm in the upper part of the mold, and then each surface of the diaphragm was propped at an angle to the direction of magnetization through the lower part of the mold. After that, the diaphragm with the mold was placed on the magnetizer (Shenzhen Shengfengyuan Automation Equipment Co. Ltd., SFY-2070) and performed by a 3 or –3 T pulse magnetic field (vertical direction), as shown in Supplementary Fig. 11c. Finally, the radial magnetization was generated as shown in Fig. 1d, e. Supplementary Movie 7 showed the magnetization process of the folded diaphragm.

### Deformation characteristics experiment

To illustrate the outstanding deformation property of the proposed magnetic folded diaphragm in the small magnetic field, compared with the magnetic flat diaphragm, the deformation properties of folded and flat diaphragms with the same magnetization process and bottom area were recorded by the camera, and the test platform was shown in Supplementary Fig. 13. An electromagnet (Dexing Magnet Tech. Co. Ltd., DXSBX-90) was utilized to generate the required test signals with a Gauss magnetometer (Dexing Magnet Tech. Co. Ltd., DX-103, range: ±200 mT, 1% FS) to monitor the magnetic field. The magnetic intensity ranged from 0 to 40 mT with 5 mT increments and the direction of the magnetic field was vertical. A digital camera (Canon EOS M6MK2) was used to record the deformation properties of diaphragm samples under different magnetic field conditions, and the displacement of the diaphragm in the top point was identified through image recognition technology. The front-view photos of the diaphragms were shown in Supplementary Fig. 14. In order to verify the reproducibility of the deformation property subjected to the low magnetically driving field, each sample was tested three cycles. The results and standard deviations are shown in Fig. 2c and Supplementary Figs. 3–8.

### Elastic-resistance force test

In order to compare the elastic-resistance force between the proposed folded and flat magnetic diaphragms, an elastic-resistance force test platform was designed based on the TA ElectroForce system (ElectroForce 3200 series, TA Instruments, BOSE) together with a 22 N load cell (0.2% FS), which could provide the gram-force load's test, as shown in Supplementary Fig. 15. The displacement data was measured by the integrating displacement sensor (range: ±7.5 mm, 0.03% FS). The flat and proposed folded diaphragms with different shapes were bound in the PLA outer frame through the glue and then installed in the load cell through the PLA basic. A non-magnetic press-head with 2 mm diameter end rounding was installed in the top actuator. The press head forced the center of the diaphragm from 0 to 10 mm with a 1 mm increment, and the force and displacement data were recorded by the TA 3200 ElectroForce controller. Each sample was tested in three cycles. The test results are shown in Fig. 2d.

### Performance test of diaphragm pump

The single diaphragm pump consisted of one Type+ folded diaphragm, two magnetic sheets, two PLA channels, and one PLA pump body, and its dimensions were shown in Supplementary Fig. 9a. The

magnetic sheets were attached to the inside and outside walls of the pump body, respectively, which could cover the pump inlet and outlet. After that, the folded diaphragm and PLA channels were glued to the pump body. Compared with the single diaphragm pump, the double diaphragm pump added one Type– folded diaphragm in the bottom of the pump body, as presented in Supplementary Fig. 9b. Their weights were measured by the electronic balance (JE302, Shanghai Puchun Measuring Instruments Co., range: $300 \pm 0.1$ g), and the results are shown in Supplementary Fig. 10.

A custom LabVIEW program (Version 2014, 32-bit), a data acquisition board (Model PCI-6229, National Instruments), a linear power amplifier (AE Techron 7224), and an electromagnet (Dexing Magnet Tech. Co. Ltd., DXSBX-80) were used to generate the harmonic magnetic field and record the measurement data for the performance tests of the proposed diaphragm pumps. The pressure and flow rate test platforms were shown in Supplementary Fig. 16.

For the pressure test, two manometers (XGZP6847A Pressure sensor modules, CFSensor, range: –5 to 5 kPa, 1% FS), which measured the pump-in and pump-out pressures, were installed in the inlet and outlet pipes (4 mm internal diameter), as shown in Supplementary Fig. 16a. To avoid stress relaxation effect of the elastomer, the pressure performance of diaphragm pump was evaluated under the harmonic magnetic field (1 Hz frequency with amplitudes (5, 10, 15, 20, 30, 40 mT) and vertical direction), and the maximum pump-in and pump-out pressures under different magnetic field amplitude conditions were shown in Fig. 3i.

For the ideal flow rate test, the magnetic sheets to achieve check valve function were removed and the outlet port was closed. Two single directional flow rate sensors (SIARGO FS4001-500-CV-A, range:0–500 mL min$^{-1}$, 1% FS), connected to the inlet port of the pump through the rubber tube (4 mm internal diameter), were used to measure the instantaneous inlet and outlet flow rate, as shown in Supplementary Fig. 16b, and then the ideal cumulative flow rate was calculated based on harmonic magnetic field input with 10 s duration. In this test, the harmonic magnetic field (vertical direction) was performed in single and double diaphragm pumps with six amplitude sets (5, 10, 15, 20, 30, 40 mT) subjected to four frequency sets (1, 2, 5, 10 Hz). The ideal flow rate for each magnetic field condition was performed three times. The results are shown in Fig. 3j, k.

The last test was the unidirectional fluid transfer function. Two water accumulators (instead by the 2 mL syringe body) were installed in the inlet and outlet ports, and the water was filled with half of the cavity (about 1 mL). A single diaphragm pump was performed in a –40 to 40 mT harmonic magnetic field with 1 Hz frequency (vertical direction), and the result was shown in Fig. 3l and Supplementary Movie 1. The double diaphragm pump was performed in –20 to 20 mT harmonic magnetic field with 5 Hz frequency (vertical direction), and the result is shown in Fig. 3m.

### Test of bionic crawling robot

The single section of the bionic crawling robot (total length: 21.6 mm, body diameter: 22 mm) consisted of two-folded diaphragms (different magnetizations, their size was shown in Supplementary Fig. 12d), 16 pieces of magnetic sheets with thickness direction magnetization (their dimensions were identical to the magnetic sheets used in diaphragm pump, as shown in Supplementary Fig. 9), eight wheels (diameter: $7 \times 3$ mm, thickness: 2 mm), one head body and one tail body (diameter: $22 \times 18$ mm, height: 8 mm). Eight magnetic sheets were glued to the head body, and their magnetization direction was opposite to the robot's head. Eight magnetic sheets were glued to the tail body, and their magnetization direction was positive to the robot's head. The head and tail bodies were equipped with four nylon wheels, respectively, in order to provide support and release the crawling resistance. After gluing the diaphragms, the head and tail parts were connected by the PLA connector. Compared with the single section of the bionic crawling robot, the double sections of the bionic crawling

robot (total length: 27 mm, body diameter: 22 mm) added one Type+ and one Type− folded diaphragms, connected by the PLA ring through the glue. For the first series experiment, four types of channels were utilized to show the ability of horizontal crawling, climbing, turning, and over the triangular slope, and the results are shown in Supplementary Movie 2. Here, the electromagnet generated −40 to 40 mT harmonic homogeneous magnetic field with horizontal direction, which was generated by the magnetic field generator utilized in the performance test of the diaphragm pump. The second series experiment illustrated the effect of different harmonic magnetic fields on the locomotion properties of the crawling robot. The applied harmonic magnetic fields were 1 Hz−±40 mT, 2 Hz−±40 mT, 4 Hz−±40 mT, 8 Hz−±40 mT, 1 Hz−±30 mT, and 1 Hz−±20 mT, respectively, with the horizontal direction. The results are shown in Supplementary Movie 3. For the third series experiment, the crawling robot showed the long-distance crawling ability in the pipe channel (length: 150 mm, internal diameter: 30 mm, external diameter: 34 mm) driven by the solenoid magnetic field generator (65 mm external diameter, 35 mm internal diameter, 140 mm length, 1 mm wire diameter, 1500 turns), and the frequency and amplitude of the harmonic magnetic field were 2 Hz and ±40 mT, respectively. The test results are shown in Supplementary Movie 4.

### Test of bionic swimming robot

The bionic swimming robot (total length: 59 mm, body diameter: 22 mm) consisted of one Type+ folded diaphragm (its size was shown in Supplementary Fig. 12d), 10 pieces of magnetic sheets with thickness direction magnetization, five pieces of magnetic sheets with length-wise magnetization (their dimensions were similar to the magnetic sheets used in diaphragm pumps, as shown in Supplementary Fig. 9), one jet pipe (diameter: 4 × 2 mm, length: 10 mm), one PLA headshell (Conical, bottom diameter: 20 mm, height: 40 mm, thickness: 1 mm), and one PLA cavity body (external diameter: 22 mm, height: 10 mm), as shown in Fig. 5 and Supplementary Fig. 2e. These components were assembled by glue. The water jet phenomenon and bionic tentacles' motions of the swimming robot without the headshell were shown in Fig. 4e, f, and Supplementary Movie 5. For this experiment, the harmonic magnetic field was generated by the magnetic field generator utilized in the performance test of the diaphragm pump. The frequency and amplitude of the harmonic magnetic field were 1 Hz and ±40 mT, respectively, and the direction was vertical direction. Besides, a solenoid magnetic field generator (65 mm external diameter, 35 mm internal diameter, 140 mm length, 1 mm wire diameter, 1500 turns) was utilized to provide the long swimming channel (length: 150 mm, internal diameter: 30 mm, external diameter: 34 mm) and −40-40 mT harmonic magnetic field subjected to 2 Hz frequency, which was driven by a power amplifier (AE Techron 7224) and monitored by a gaussmeter. The direction of the magnetic field was vertical in the snorkeling and diving tests, and it is horizontal in the horizontal diving test. The snorkeling, diving, and horizontal diving of the bionic swimming robot were shown in Supplementary Movie 6.

## Data availability

All data supporting the findings of this study are available within the article and the Supplementary Information file, or available from the corresponding authors upon request. The data generated in this study are provided in the Source Data file. Source data are provided with this paper.

## Code availability

Codes are available from the corresponding author upon request.

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

## Acknowledgements

F.Y. acknowledges the support from the National Natural Science Foundation of China (Grant numbers 61733006 and U1813201).

## Author contributions

D.L., F.Y., and D.G. conceived the idea and led research efforts. D.L., D.G., and R.L. performed the experiments. D.L. and F.Y. wrote the manuscript with the assistance of other co-authors.

## Competing interests

The authors declare no competing interests.
