## [Peer Review File · Nature Communications]

Bio-inspired magnetic-driven folded diaphragm for biomimetic robotREVIEWER COMMENTS

Reviewer #1 (Remarks to the Author):

This work reports a type of one-piece molded folding diaphragm with radial magnetization, and three types of applications such as diaphragm pump, the crawling robot, and the swimming robot are designed. This is an interesting soft robot approach. However, considering the high requirement of the Nature Communication, the paper has many weakness and the experimental tests are too simple and incomplete. It is unsuitable for publication in Nature Communication in this current form. In my opinion, this work requires a major revision.

Detailed comments are given below:

1. As many soft robots using magnetic fields have been proposed and they seem to be more responsive. Meanwhile, origami structure has applications in soft robots. We recommend that the innovation point and advantages of using the folding diaphragm are mentioned in the introduction.
2. In Fig. 2, when comparing the deformation range between the folding and flat magnetic diaphragm in triangle, square, hexagon, and circular, more comprehensive tests should be set up to comprehensively study the deformation mechanism of folding diaphragm. For example, the folding magnetic diaphragm with different thicknesses, different folding degrees, different number of segments, different hardness of the silicone elastomer, different mass ratios of NdFeB particles to Ecoflex 00-20.
3. In addition, Fig. 2 is an important diagram supporting the mechanism of this paper, but it has not been analysed by simulation or other means. It needs to be supported by comparison between actual photographs and simulations.
4. The authors mention the large deformation capability subjected to the low magnetically driving field. But the Fig. 2 just shows one cycle, How good is the reproducibility?
5. The description of the diaphragm pump seems to be lacking. A description of the size of the pump, the type of syringe, etc. is required.
6. Fig. 4 is an important diagram supporting the mechanism of this paper, but it is mainly a conceptual diagram and has not been analysed by simulation or other means. It needs to be supported by comparison between actual photographs and simulations.
7. The author mention the large deformation capability subjected to the low magnetically driving field, how good is the reproducibility?
8. There is no description of the bionic crawling robot. What is the length of the robot? The diameter?
9. A description of the magnetic field generated is required.
10. The authors should give more analysis of the results of the tests, rather than just stating the test results only.
11. In the test of the robot crawling, apart from the tests in the paper, more tests should be conducted, such as different magnetic field intensity, frequency, and direction.

Reviewer #2 (Remarks to the Author):

Noteworthy results:

- Using a folded diaphragm for soft robots that is magnetically driven,
- folded diaphragms as a mechanical "amplifier" for the resulting forces and deflections
- realization of such diaphragms with different geometry and comparison of them with respect to deflection and force
- application to biomimetic demonstrators like a crawling and a swimming robot

Significance:

- novel approach
- published results are complementary to other driving mechanisms like electroactive polymers
- magnetic excitation allows to avoid electrical connectors in particular in water and for corresponding applications

Conclusions and Claims:

- are logically and consequentially derived from the representations

Flaws:

- The title is unfortunate, since the authors are talking about folded membranes, not folding membranes. Furthermore, it should read "magnetic-driven" or "magnetically driven".
- The authors do not provide any geometric dimensions for their structures, so there is no third party repeatability or verifiability.
- The mechanical model in Supplementary information 1 remains rather unclear. Is the membrane flexurally constrained at the edge? Can the individual segments be considered ideally stiff?
- The mathematical equations (S1) to S5 are in no way suitable for understanding the operation of the folding diaphragm.
- Four different configurations are considered in Figure 2. Information about the dimensions is also completely missing. The figures for displacement and force are scaled completely differently and are therefore difficult to compare.
- In Figure 2, it would also make sense to indicate relative forces and displacements.

- The deflection of the diaphragms depends on "wave height" and "wave length". No statements are made about this.
- The magnetization process is described unclearly and poorly comprehensible. Supplementary Figure 1 does not help here either for the understanding.
- The information on the "elastic-resistance force test" device (Supplementary Figure 3) is completely inadequate. Also in the text in section 4.4. only "TA 3230 electric force system" is mentioned. Information on the manufacturer, the measuring ranges and the measurement uncertainty is completely missing. A Google search did not yield any results, so that this part is not comprehensible to me.
- In Figs. 2c,d and 3i-k, information on the measurement uncertainties is missing. Are these single measurements on a single sample?
- Information on measurement uncertainties is totally missing in the whole manuscript. Accuracy data in the text of e.g. 178.6 mL/min, 166.67 kPakg-1, 60066 ml min-1 kg-1 or a "159% higher" deformation are completely implausible.

Reviewer #3 (Remarks to the Author):

In this manuscript, Yang et al. reported on a magnetically operable soft actuator manufactured in a one-piece mold, which exhibited large, three-dimensional deformation by in response to an external magnetic field with low intensity (~40 mT). This actuator, composed of a mixture of magnetic particles and silicone rubber, was cured in a mold with a folded disk shape and then magnetized in the vertical direction with stretching the disk into a convex form. These processes radially oriented the magnetic moments, so that the actuator could undergo reversible stretching and contraction with turning on and off the vertical magnetic field. The stretching direction could be switched in upward and downward with tuning the magnet direction in upward and downward, respectively, where the folding arrangement enable large deformation with a weak magnetic field. Based on this actuation, various devices, such as gas pump, liquid pump, crawling robot, and swimming robot, were developed.

My first impression after reading the manuscript is that this is a high-quality work suitable for publication in specialized journal in robotics and engineering. All devices were carefully elaborated to realize good performances, but the material composition (NdFeB magnetic particle / silicone rubber) and fabrication method (magnetization under elastic deformation) are just an extension of previous works (e.g. ref. 29).

However, I also felt that this work implies one future direction of the field of soft actuators. Although many kinds of magnet-driving soft actuators have been reported, this is the first example of realizing the 'inside-volume change' of the actuator; all other examples achieved only the changes in lengths, shapes, and/or angles. Such volume change is the origin of the high performance of the gas/liquid pumps and the swimming robot. In my opinion, the true

novelty of this work is not the better mimicking of creatures' locomotion mechanism, but the realization of 'inside-volume change', which would expand the scope of soft actuators.

Therefore, I don't recommend the current manuscript for publication in the current form, but if it is properly property revised, I think it would become suitable for publication in this journal. Followings are the points to be addressed through the revision.

(1) In the introductory part, the difference between the present actuator and conventional magnet-driving soft actuators should be described more clearly.

(2) Diaphragm pumps based on other soft actuators have been reported, most of which are based on dielectric elastomers. The characteristics of the present magnet-driving pump compared with conventional soft actuator-based pumps should be clarified.

(3) If the design principle for controlling/maximizing the performance of the present diaphragm pump (number of folding repetitions, modulus/thickness of rubber, height of the magnetization mold) is provided, it would be helpful for the readers who want to follow the present strategy.

(4) Minor points:

(i) Section 3 conclusion: The terms "specific pressure" and "specific flow rate" should be defined.

(ii) Section 2.1.1: "Fig. 5" should read "Fig. 4".

(iii) Section 2.4.2: "Fig. 4" should read "Fig. 5".

(iv) Figure 4: There are two "(b)".

(v) Figure 2a, b and Figure 3b–d, f–h : These pictures seem to be edited by some graphic software for easy understanding. However, from scientific and ethics viewpoints, such image editing should be minimized, or ideally should not be done.

(vi) Throughout the manuscript: The authors may well be conscious about the house style commonly used in scientific journals, in terms of the use of past/present/future tenses, upper/lower-case letters, space insertion between values and units, etc.

Responses to reviewers' comments

Manuscript ID: NCOMMS-22-28657

Original Article Title: “Bio-inspired magnetic driving folding diaphragm for biomimetic robot”

Revised Article Title: “Bio-inspired magnetic-driven folded diaphragm for biomimetic robot”

Dear Editors, Nature Communications

Thank you for reconsidering for publication of our manuscript, with an opportunity to address the reviewers' comments.

We are uploading the revised manuscript, and our point-by-point response to the reviewers' comments.

Best regards,

Prof. Fan Yang

Huaqiao University, Xiamen, P.R. China

2022.12.01

Authors' Responses to the comments by

Reviewer #1

Manuscript ID: NCOMMS-22-28657

Original Article Title: “Bio-inspired magnetic driving folding diaphragm for biomimetic robot”

Revised Article Title: “Bio-inspired magnetic-driven folded diaphragm for biomimetic robot”

We would like to express our thanks to the Reviewer for the very constructive comments made. Please find below a point-by-point reply to each comment. The manuscript has also been revised based on the suggested comments.

General Comment: *“This work reports a type of one-piece molded folding diaphragm with radial magnetization, and three types of applications such as diaphragm pump, the crawling robot, and the swimming robot are designed. This is an interesting soft robot approach. However, considering the high requirement of the Nature Communication, the paper has many weakness and the experimental tests are too simple and incomplete. It is unsuitable for publication in Nature Communication in this current form. In my opinion ,this work requires a major revision.”*

Authors Response: The authors appreciate the general comment made by the Reviewer, and will provide response regarding the Reviewer’s comments point to point in the following part.

Reviewer’s Comment 1: *“As many soft robots using magnetic fields have been proposed and they seem to be more responsive. Meanwhile, origami structure has applications in soft robots. We recommend that the innovation point and advantages of using the folding diaphragm are mentioned in the introduction.”*

Authors Response: The authors appreciate the Reviewer’s valuable comment. As presented/summarized in the “Abstract” and the last paragraph of “**Introduction**” parts in the original manuscript, “*the innovation point and advantages of using the folding diaphragm*” is the proposed “*one-piece molded folding diaphragm with radial magnetization*” can achieve large 3-D and bi-direction deformation subjected to simple homogeneous and related low strength magnetic fields, and then realize relative large inside-volume changes with high strength, which makes this kind of folded diaphragm to be able to be easily customized to conduct different practical applications, such as the pump and soft robotic systems as soft drivers. The above properties are superior to other “*origami structure*” reported in the application of soft robotics, which can achieve the changes in lengths, shapes and/or angles.

Considering the valuable comment made by the Reviewer, the authors emphasized the “*the innovation point and advantages of using the folding diaphragm*” in the “**Introduction**” part of the revised manuscript and added relative references as:

1. “*Meanwhile, many kinds of the magnetic-driven soft actuators have been reported, including the origami structure^{22, 30, 36-38}, but most of them can only achieve the changes in shapes³⁰, angles³⁶, and/or lengths³⁸, and cannot realize the relatively large inside-volume changes with high strength, which may limit their potential practical application as soft actuators (drivers), such as the micro-pump.*” (the last 3rd sentence of the 2nd paragraph of the “**Introduction**” part of the revised manuscript)
2. “*Therefore, in this study, a kind of magnetic-driven folded diaphragm, with one-piece molded and simple manufacturing procedure, and large, 3-D and bi-direction deformation subjected to simple homogeneous and related low strength magnetic fields, will be proposed. This kind of folded diaphragm with different radial magnetization properties is able to realize large inside-volume changes with high strength, which is inspired by the locomotion of earthworm generated by the contraction and stretching between body segments.*” (the 1st and 2nd sentences of the 3rd paragraph of the “**Introduction**” part of the revised manuscript)”

Reviewer's Comment 2: *“In Fig. 2, when comparing the deformation range between the folding and flat magnetic diaphragm in triangle, square, hexagon, and circular, more comprehensive tests should be set up to comprehensively study the deformation mechanism of folding diaphragm. For example, the folding magnetic diaphragm with different thicknesses, different folding degrees, different number of segments, different hardness of the silicone elastomer, different mass ratios of NdFeB particles to Ecoflex 00-20.”*

Authors Response: The authors appreciate the Reviewer's valuable comment. Considering the Reviewer's valuable comment, the authors added **“Supplementary Discussion 3. The deformation properties of folded diaphragm”** in the revised submission, to supplement the comprehensive test and analysis to evaluate *“the deformation mechanism of folded diaphragm”* considering multiple conditions, which include the diaphragm thickness, folding degree, number of segments, hardness of the silicone elastomer, weight ratio of NdFeB particles to silicone elastomer and the magnetization degree. At the same time, the authors made necessary modification in the revised manuscript as:

(1) adding one paragraph in Sub-section 2.2 **“Mechanical properties”** just before Fig. 2 in the revised manuscript as “Supplementary Discussion 3 summarized the deformation characteristics of the magnetic-driven folded diaphragm subjected to multiple conditions, which include the diaphragm thickness, folded degree, number of segments, hardness of the silicone elastomer, weight ratio of NdFeB particles to silicone elastomer and the magnetization degree. The fabrication conditions are presented in Supplementary Table 1 and the comparison results of different fabrication conditions are shown in Supplementary Fig. 3-8.” (the last paragraph of the Sub-section 2.2 **“Mechanical properties”** in the revised manuscript)

(2) adding two sentences at the end of Sub-section 4.1 **“Material fabrication”** in the revised manuscript to illustrate the testing sample as “In this study, 14 types of prototypes (3 samples for each prototype) with different diaphragm thickness, folded degree, number of segments, hardness of the silicone elastomer, weight ratio of NdFeB particles to silicone elastomer and the magnetization degree, as summarized in Supplementary Table 1, have been fabricated and tested to investigate the

deformation and elastic-resistance force properties of the proposed folded diaphragms. Here, it should be noted that each sample (3 samples for each prototype) has been test 3 times, and then total 9 tests for each prototype, as summarized in Fig 2 and Supplementary Fig. 3-8, to verify the reproducibility of the deformation properties.” (the last two sentences at the end of Sub-section 4.1 “**Material fabrication**” in the revised manuscript)

Reviewer’s Comment 3: *“In addition, Fig. 2 is an important diagram supporting the mechanism of this paper, but it has not been analysed by simulation or other means. It needs to be supported by comparison between actual photographs and simulations.”*

Authors Response: The authors appreciate the Reviewer’s valuable comment. In the revised manuscript, the authors supplemented the comparison between experiments and simulations of the deformation properties of folded and flat diaphragms, as shown in “**Supplementary Discussion 2. Simulations of the folded diaphragm and its applications**” of the Supplementary Information.

Reviewer’s Comment 4: *“The authors mention the large deformation capability subjected to the low magnetically driving field. But the Fig. 2 just shows one cycle, How good is the reproducibility?”*

Authors Response: The authors appreciate the Reviewer’s valuable comment. In this study in order to verify the reproducibility of the large deformation capability subjected to the low magnetically driving field, 3 samples have been fabricated for each type of diaphragm, and each sample has been tested 3 cycles. Therefore, each type of diaphragm has been tested for 9 cycles totally. However, in the original manuscript, the authors did not mention the above point clearly. The authors appreciate again for the Reviewer’s valuable comment.

Considering the Reviewer’s valuable comment, the author modified the first paragraph of Sub-section 2.1 “**Mechanical properties**” as: “The deformation characteristics are one of the most important indexes to evaluate the performance of the proposed folded diaphragm. Furthermore, as mentioned before, the folded diaphragm can be customized to different shapes to meet the

requirement of practical applications. Therefore, in this section the deformation properties of triangle, square, hexagon and circular shapes of the proposed folded diaphragm, as shown in Fig. 2a, will be evaluated based on the designed experimental, and compared with the normal magnetic flat diaphragm. In this study, in order to verify the reproducibility of the large deformation capability subjected to the low magnetically driving field, 3 samples have been fabricated for each type of diaphragm shown in Fig.2a, and each sample has been tested 3 cycles. Therefore, each type of diaphragm shown in Fig.2a has been tested for 9 cycles totally. Fig. 2b clearly illustrates the outstanding deformation properties of the proposed magnetic folded diaphragm compared with those of flat one, which should overcome the large elastic-resistance force due to the extension of the flat diaphragm, under same magnetic field. For example, the deformation of circular folded diaphragm ($5.568 \pm 0.118 \text{ mm}$) is 146% larger than that of circular flat diaphragm ($2.267 \pm 0.114 \text{ mm}$) under same magnetic field, as shown in Fig. 2c, which is mainly because of the much small elastic-resistance force generated from the folded diaphragm compared with that of the flat one under the same deformation condition, as shown in Fig. 2d. The large deformation property of the folded diaphragm can be observed from the simulation result, too, as shown in Supplementary Fig. 2a, b in the Supplementary Discussion 2. Here, it should be noted that the result illustrated in Fig. 2c, d include the average value and standard deviation of 9 testing cycle for each type of diaphragm, and the maximum standard deviation, which is around $1.397 \pm 0.323 \text{ mm}$, is located in the deformation curve for the triangle folded diaphragm subjected to 15 mT magnetic field.” (the *Italic* part is the modified part).

At the same time, in the revised submission for the data shown in Fig. 2 and Supplementary Fig. 3-8, the authors provided the average value and its standard deviation based on 9 cycles measurement to illustrate the “reproducibility” properties.

Reviewer’s Comment 5: “*The description of the diaphragm pump seems to be lacking. A description of the size of the pump, the type of syringe, etc. is required.*”

Authors Response: The authors appreciate the Reviewer’s valuable comment. “*The description of the diaphragm pump*” was summarized in 2nd and 3rd sentences in the 1st paragraph of Sub-section 2.3 “**Implementation in diaphragm pump**” in the original manuscript. Considering the Reviewer’s valuable comment,

(1) In Sub-section 4.5 “**Performance test of diaphragm pump**” of the revised manuscript, the authors added one paragraph to provide the detailed description of the diaphragm pump as “*The single diaphragm pump consists of one Type + folded diaphragm, two magnetic sheets, two PLA channels and one PLA pump body, and its dimensions is shown in Supplementary Fig. 9a. The magnetic sheets are attached to the inside and outside walls of pump body, respectively, which can cover the pump inlet and outlet. After that, the folded diaphragm and PLA channels are glued on the pump body. Compared with the single diaphragm pump, the double diaphragm pump adds one Type - folded diaphragm in the bottom of pimp body, as presented in Supplementary Fig. 9b. Their weights are measured by the electronic balance (JE302, Shanghai Puchun Measuring Instruments Co., range: 300 g ± 0.1 g), and the results is shown in Supplementary Fig. 10.*”

(2) In the revised submission, the authors added Supplementary Fig. 9 and 10 to illustrate the detailed dimension and the weight of the proposed diaphragm pumps.

Supplementary Figure 9. **The size information of diaphragm pump. a** The single diaphragm pump. **b** The double diaphragm pump

Supplementary Figure 10. **The weight of diaphragm pump.** **a** Single diaphragm pump. **b** double diaphragms pump.

Besides, the “syringe” is just instead of the water accumulator and shows the liquid level through the tick mark of the syringe. In the revised manuscript, the authors add the dimension of the syringe, as *“Two water accumulators (instead by the 2mL syringe body) are installed in the inlet and outlet ports, and the water is filled with half of cavity (about 1 mL).”* (The beginning of the last paragraph of Sub-section 4.5 **“Performance test of diaphragm pump”** in the revised manuscript)

Reviewer’s Comment 6: *“Fig. 4 is an important diagram supporting the mechanism of this paper, but it is mainly a conceptual diagram and has not been analysed by simulation or other means. It needs to be supported by comparison between actual photographs and simulations.”*

Authors Response: The authors appreciate the Reviewer’s valuable comment. In the revised manuscript, the authors supplemented the comparison between actual photographs and simulations of the deformation properties of diaphragm pump, crawling robot and swimming robot, as shown in **“Supplementary Discussion 2. Simulations of the folded diaphragm and its applications”** of the Supplementary Information.

Reviewer’s Comment 7: *“The author mention the large deformation capability subjected to the low magnetically driving field, how good is the reproducibility?”*

Authors Response: The authors appreciate the Reviewer’s valuable comment. As presented in the response of the Reviewer’s Comment 4, 3 samples have been fabricated for each type of diaphragm, and each sample has been tested 3 cycles. Therefore, each type of diaphragm has been tested for 9 cycles totally. The reproducibility has been presented in Fig. 2 in the revised manuscript. Furthermore, the data to support the response for the Reviewer’s Comment 2, as presented in Supplementary Fig. 3-8, was also based on the 9 cycles test. The detailed information regarding this valuable comment has been provide in the response of Reviewer’s Comments 2 and 4.

Reviewer’s Comment 8: *“There is no description of the bionic crawling robot. What is the length of the robot? The diameter?”*

Authors Response: The authors appreciate the Reviewer’s valuable comment. In the revised manuscript, the authors presented the dimensions of the bionic crawling robot, as: “The single section of bionic crawl robot (totally length: 21.6 mm, body diameter: 22 mm) consists of two folded diaphragms (different magnetizations, its size is shown in Supplementary Fig. 12d), sixteen pieces of magnetic sheets with thickness direction magnetization (their dimensions are identical to the magnetic sheets used in diaphragm pump), eight wheels (diameter: 7×3 mm, thickness: 2 mm), one head body and one tail body (diameter: 22×18 mm, height: 8 mm). Eight magnetic sheets are glued to the head body, and their magnetization direction is opposite to the robot's head. Eight magnetic sheets are glued to the tail body, and their magnetization direction is positive to the robot's head. The head and tail bodies are equipped with four nylon wheels, respectively, in order to provide support and release the crawl resistance. After gluing the diaphragms, the head and tail parts are connected by the PLA connector. Compared with the single section of bionic crawl robot, the double sections of bionic crawl robot (totally length: 27 mm, body diameter: 22 mm) adds one Type+ and on Type-

folded diaphragms, connected by the PLA ring through the glue.” (at the beginning of the 1st paragraph of Sub-section 4.6 “**Bionic crawling robot.**”, the *Italic* part is the modified part)

Besides, the size of the swimming robot is also supplemented, as: “The bionic swimming robot (totally length: 59 mm, body diameter: 22 mm) consists of one Type+ folded diaphragm (its size is shown in Supplementary Fig. 12d), ten pieces of magnetic sheets with thickness direction magnetization, five pieces of magnetic sheets with lengthwise magnetization (their dimensions are same as the magnetic sheets used in diaphragm), one jet pipe (diameter: 4×2 mm, length: 10 mm), one PLA head shell (Conical, bottom diameter: 20 mm, height: 40 mm, thickness: 1 mm) and one PLA cavity body (external diameter: 22 mm, height: 10 mm), as shown in Fig.5 and Supplementary Fig. 2e.” (at the beginning of the 1st paragraph of Sub-section 4.7 “**Bionic swimming robot**”, the *Italic* part is the modified part).

Reviewer’s Comment 9: “A description of the magnetic field generated is required”

Authors Response: The authors appreciate the Reviewer’s valuable comment. Considering the valuable comment from the Reviewer, the author makes the following modifications in the revised manuscript to provide the detailed description of the magnetic field utilized in this study:

(a) the authors added the brand of the magnetizer utilized in the procedure of the diaphragm magnetization as: “After that, the diaphragm with the mold is placed on the magnetizer (Shenzhen Shengfengyuan Automation Equipment Co. Ltd., SFY-2070) and performed by a 3 T or -3 T pulse magnetic field (vertical direction), as shown in Supplementary Fig. 11c. Finally, the radial magnetization is generated as shown in Fig. 1d, e. Supplementary Movie 7 shows the magnetization process of the folded diaphragm.” (at the end of Sub-section 4.2 “**Diaphragm magnetization**” in the revised manuscript.)

(b) the authors added the detailed data for the magnetic field utilized for the deformation characteristics experiment as: “The magnetic intensity ranges from 0 to 40 mT with 5 mT increment

and the direction of the magnetic field is vertical.” (the 3rd sentence of Sub-section 4.3 “**Deformation characteristics experiment**” in the revised manuscript).

(c) the vertical direction of the magnetic field in the diaphragm pump test is supplemented in the Sub-section 4.5 “**Performance test of diaphragm pump**”.

(d) the magnetic field used in the long-distance crawling test is described as: “For the third series experiment, the crawling robot shows the long-distance crawling ability in the pipe channel (length: 150 mm, internal diameter: 30 mm, external diameter: 34 mm) driven by the solenoid magnetic field generator (65 mm external diameter, 35 mm internal diameter, 140 mm length, 1mm wire diameter, 1500 turns), and the frequency and amplitude of the harmonic magnetic field are 2 Hz and ± 40 mT, respectively.” (the last sentence of Sub-section 4.6 “**Bionic crawling robot**” in the revised manuscript, and the *Italic* part is the modified part)

(e) the magnetic field used in the swimming robot are described as: “For this experiment, the harmonic magnetic field is generated by the magnetic field generator utilized in performance test of diaphragm pump (Subsection 4.5). The frequency and amplitude of the harmonic magnetic field are 1 Hz and ± 40 mT respectively, and the direction is vertical direction. Besides, a solenoid magnetic field generator (65 mm external diameter, 35 mm internal diameter, 140 mm length, 1 mm wire diameter, 1500 turns) is utilized to provide the long swimming channel (length: 150 mm, internal diameter: 30 mm, external diameter: 34 mm) and -40~40 mT harmonic magnetic field subjected to 2 Hz frequency, which is driven by a power amplifier (AE Techron 7224) and monitored by a gaussmeter. The direction of the magnetic field is vertical in the snorkeling and diving test, and it is horizontal in horizontal diving test.” (The 3rd sentence of Sub-section 4.7 “**Bionic swimming robot**” in the revised manuscript, the *Italic* part is the modified part)

Reviewer’s Comment 10: *“The authors should give more analysis of the results of the tests, rather than just stating the test results only.”*

Authors Response: The authors appreciate the Reviewer’s valuable comment. In the revised manuscript, the authors added (1) the detailed diaphragm test samples and their deformation properties to illustrate the properties of reproducibility, as presented for the reviewers comment 4 and 7; (2) the detailed testing procedure/result for the diagram with different dimension, as presented for the reviewer’s comment 2; (3) the comparison of the deformation of the proposed folded diagram including its application is soft robot between the simulation result and the test data has been provided in the revised submission, as presented for the reviewer’s comments 3 and 6; (4) the detailed pumping dimension, as presented for the reviewer’s comment 5, and the pumping performance, as added in the Supplementary Figures 14, and Supplementary Tables 2- 4.

Reviewer’s Comment 11: *“In the test of the robot crawling, apart from the tests in the paper, more tests should be conducted, such as different magnetic field intensity, frequency, and direction.”*

Authors Response: The authors appreciate the Reviewer’s valuable comment. Considering the valuable comment from the Reviewer, the authors makes the following modifications:

(a) the authors stated the dependency of magnetic field conditions on the locomotion properties of the crawling robot, as: *“Meanwhile, the locomotion speed of the crawling robot can be modified through different magnetic field intensities and frequencies, as shown in Supplementary Movie 3.”*

(at the end of 2nd paragraph in the Sub-section 2.4 “**Bionic earthworm robots**” of the revised manuscript)

(b) the authors presented the effect of the magnetic fields on the locomotion properties of the crawling robot, as: *“The second series experiment illustrated the effect of different harmonic magnetic fields on the locomotion properties of the crawling robot. The applied harmonic magnetic fields are 1Hz-±40mT, 2Hz-±40mT, 4Hz-±40mT, 8Hz-±40mT, 1Hz-±30mT and 1Hz-±20mT, respectively, with”*

horizontal direction. The results are shown in the Supplementary Movie 3.” (in the middle of Sub-section 4.6 “**Bionic crawling robot**” in the revised manuscript)

(c) the authors supplemented the video of the crawling robot under different magnetic field conditions in the Supplementary Movie 3.

The authors would like to express their appreciations again to you for the time and effort you devoted to review our manuscript and to provide us with your constructive suggestions, which we have fully implemented in the revised manuscript.

With sincere appreciation,

Prof. Fan Yang
Huaqiao University, Xiamen, P.R. China
2022.12.01

Authors' Responses to the comments by

Reviewer #2

Manuscript ID: NCOMMS-22-28657

Original Article Title: "Bio-inspired magnetic driving folding diaphragm for biomimetic robot"

Revised Article Title: "Bio-inspired magnetic-driven folded diaphragm for biomimetic robot"

We would like to express our thanks to the reviewer for the very constructive comments made. Please find below a point-by-point reply to each comment. The manuscript has also been revised based on the suggested comments.

General Comment: *"Noteworthy results:*

- *Using a folded diaphragm for soft robots that is magnetically driven,*
- *folded diaphragms as an mechanical "amplifier" for the resulting forces and deflections*
- *realization of such diaphragms with different geometry and comparison of them with respect to deflection and force*
- *application to biomimetic demonstrators like a crawling and a swimming robot*

Significance:

- *novel approach*
- *published results are complementary to other driving mechanisms like electroactive polymers*

- magnetic excitation allows to avoid electrical connectors in particular in water and for corresponding applications

Conclusions and Claims:

- are logically and consequentially derived from the representations”

Authors Response: The authors appreciate the general comment made by the Reviewer, and will provide response regarding the reviewer’s comments point to point in the following part.

Reviewer’s Comment 1: “*The title is unfortunate, since the authors are talking about folded membranes, not folding membranes. Furthermore, it should read "magnetic-driven" or "magnetically driven".*”

Authors Response: The authors appreciate the reviewer’s valuable comment. Considering the valuable comment from the reviewer, the title is modified as “Bio-inspired magnetic-driven folded diaphragm for biomimetic robot”

Reviewer’s Comment 2: “*The authors do not provide any geometric dimensions for their structures, so there is no third party repeatability or verifiability.*”

Authors Response: The authors appreciate the reviewer’s valuable comment. Considering the reviewer’s valuable comment, in the revised Supplementary Information, the authors provided the detailed information of the geometric dimensions of structures, which include different folded diaphragms presented in Fig. 2, and pumps as:

(a) Folded diaphragms

Supplementary Figure 12. **The size information of four configurations. a Triangle. b Square. c Hexagon. d Circular.**

(b) Pumps

Supplementary Figure 9. **The size information of diaphragm pump. a The single diaphragm pump. b The double diaphragm pump**

And then, the authors made necessary modification in the revised main manuscript as:

(a) “Due to different requirements of the actuator shape, the diaphragms (folded and flat) with different shapes (triangle, square, hexagon and circular) are fabricated (as shown in Supplementary Fig. 11a), which are used to verify the outstanding deformation property of the magnetically folded diaphragm compared with the flat diaphragm (their dimensions are shown in Supplementary Fig. 12).” (at the end of the 4th sentence in Sub-section 4.1 **“Material fabrication”**)

(b) “The detailed dimensions of both pumps are provided in Supplementary Fig. 9, and its weights are shown in Supplementary Fig. 10.” (the 6th sentence of 1st paragraph in Sub-section 2.3 **“Implementation in diaphragm pump”**)

At the same time, the geometric dimension of bionic crawl robot and crawling robot have been added in the revised main manuscript as:

(1) “The single section of bionic crawl robot (totally length: 21.6 mm, body diameter: 22 mm) consists of two folded diaphragms (different magnetizations, its size is shown in Supplementary Fig. 12d), sixteen pieces of magnetic sheets with thickness direction magnetization (their dimensions are identical to the magnetic sheets used in diaphragm pump), eight wheels (diameter: 7×3 mm, thickness: 2 mm), one head body and one tail body (diameter: 22×18 mm, height: 8 mm).” (at the beginning of the 1st paragraph in Sub-section 4.6 **“Bionic crawling robot”**, the *Italic* part is the modified part);

(2) “Compared with the single section of bionic crawl robot, the double sections of bionic crawl robot (totally length: 27 mm, body diameter: 22 mm) adds one Type+ and one Type- folded diaphragms, connected by the PLA ring through the glue” (at the beginning of Sub-section 4.6 **“Bionic crawling robot.”**);

(3) “The bionic swimming robot (totally length: 59 mm, body diameter: 22 mm) consists of one Type+ folded diaphragm (its size is shown in Supplementary Fig. 12d), ten pieces of magnetic sheets with thickness direction magnetization, five pieces of magnetic sheets with lengthwise magnetization (their dimensions are same as the magnetic sheets used in diaphragm), one jet pipe (diameter: 4×2 mm, length: 10 mm), one PLA head shell (Conical, bottom diameter: 20 mm, height: 40 mm, thickness: 1 mm) and one PLA cavity body (external diameter: 22 mm, height: 10 mm), as shown in Fig.5 and Supplement Fig. 2e.” (at the beginning of the 1st paragraph in Sub-section 4.7 **“Bionic swimming robot”**, the *Italic* part is the modified part).

Reviewer’s Comment 3: *“The mechanical model in Supplementary information 1 remains rather unclear. Is the membrane flexurally constrained at the edge? Can the individual segments be considered ideally stiff?”*

Authors Response: The authors appreciate the Reviewer’s valuable comment. It is true that in this study, the individual segments have been considered as ideally stiff, and connected by torsion spring at the edge. Considering the reviewer’s valuable comment, the above point has been emphasized in the revised supplement information as: “Due to the folding structure of the circular diaphragm shown in Fig. S1(a), the deformation mechanism of magnetic diaphragm in each segment can be assumed (simplified) as a multi-link mechanism with rotation joint (J_i) located at each segment end point, as shown in Fig. S1(b), and the individual segments have been considered as ideally stiff, and connected by torsion spring at the edge.” (the 1st paragraph in Supplementary Discussion 1, the *Italic* part is the modified part)

Reviewer’s Comment 4: *“The mathematical equations (S1) to S5 are in no way suitable for understanding the operation of the folding diaphragm.”*

Authors Response: The authors appreciate the Reviewer’s valuable comment. The authors have re-organized the **Supplementary Discussion 1** to illustrate the operation principle of the proposed folded diaphragm in the revised submission.

Reviewer’s Comment 5: *“Four different configurations are considered in Figure 2. Information about the dimensions is also completely missing. The figures for displacement and force are scaled completely differently and are therefore difficult to compare.”*

Authors Response: The authors appreciate the Reviewer’s valuable comment. In the original manuscript, the authors missed the detailed dimension of folded diaphragms. The authors appreciate again for the Reviewer’s valuable comment. In the revised manuscript, the authors provide detailed

configurations information of four different types of folded diaphragms in the Supplementary Fig. 12, as:

Here, it should be noted that the authors kept at least the dimension of one side of the Triangle, Square and Hexagon samples to be 20mm, which is the radius of the circular sample, to compare their performance.

Furthermore, the authors made the following modification in the revised manuscript based on the Reviewer' valuable comment.

(a) In the revised manuscript, (1) Fig. 2a,b has been scaled by same ratio, and the scale has been marked in the Fig. 2a,b; (2) the axis in the Fig. 2c,d has been scaled identically.

Figure 2. **Mechanical properties of the folded diaphragm.** **a** Customizable shape of magnetic diaphragm in triangle, square, hexagon and circular. **b** comparisons of deformation range between the folded and flat magnetic diaphragm under the 40mT magnetic field by vision. **c** comparisons of the displacement (average value and standard deviation of 9 cycles measurement) in top point between the folded and flat magnetic diaphragm under different magnetic fields (range from 0-40mT with 5mT increment). **d** comparisons of the elastic force of diaphragm center between the folded and flat magnetic diaphragm under different displacements ranging from 0-10mm with 1mm increment (average value and standard deviation of 9 cycles measurement).

(b) the authors added the front-view photo of the different configurations under different magnetic fields in the Supplementary Fig. 14, as:

Magnetic field	Triangle		Square		Hexagon		Circular	
	Flat	Folding	Flat	Folding	Flat	Folding	Flat	Folding
0 mT	10 mm		10 mm		10 mm		10 mm	
5 mT								
10 mT								
15 mT								
20 mT								
25 mT								
30 mT								
35 mT								
40 mT								

Supplementary Figure 14. The comparisons of diaphragm deformation of four configurations (triangle, square, hexagon, circular) in front view.

Reviewer’s Comment 6: “In Figure 2, it would also make sense to indicate relative forces and displacements.”

Authors Response: The authors appreciate the reviewer’s valuable comment. As presented in the Response of Reviewer’s Comment 5, in the revised manuscript, the authors provided detailed dimension of the diaphragm, and kept at least the dimension of one side of the Triangle, Square and Hexagon samples to be 20mm shown in Supplementary Fig. 12, which is the radius of the circular sample, to compare their performance.

Reviewer’s Comment 7: “The deflection of the diaphragms depends on “wave height” and “wave length”. No statements are made about this.”

Authors Response: The authors appreciate the Reviewer’s valuable comment. It is true that “The deflection of the diaphragms depends on “wave height” and “wave length”.”, which depend on the folded degree under the similar boundary condition of the diaphragm. The large folded degree results large length and height of the diaphragm. Considering the valuable comment of reviewer, the author compares the deformation properties between different folding degree (35°, 45°, 55°), and states the

effect of the "wave height" and "wave length" on the deformation properties of diaphragms, as: *“Supplementary Discussion 3 summarized the deformation characteristics of the magnetic-driven folded diaphragm subjected to multiple conditions, which include the diaphragm thickness, folded degree, number of segments, hardness of the silicone elastomer, weight ratio of NdFeB particles to silicone elastomer and the magnetization degree. The fabrication conditions are presented in Supplementary Table 1 and the comparison results of different fabrication conditions are shown in Supplementary Fig. 3-8.”* (the last paragraph of the Sub-section 2.2 “**Mechanical properties**” in the revised manuscript)

Reviewer’s Comment 8: *“The magnetization process is described unclearly and poorly comprehensible. Supplementary Figure 1 does not help here either for the understanding.”*

Authors Response: The authors appreciate the Reviewer’s valuable comment. Considering the reviewer’s valuable comment, the magnetization process has been applied in Supplementary Movie 7.

Reviewer’s Comment 9: *“The information on the "elastic-resistance force test" device (Supplementary Figure 3) is completely inadequate. Also in the text in section 4.4. only "TA 3230 electric force system" is mentioned. Information on the manufacturer, the measuring ranges and the measurement uncertainty is completely missing. A Google search did not yield any results, so that this part is not comprehensible to me.”*

Authors Response: The authors appreciate the Reviewer’s valuable comment. TA 3230 electric force system belongs to the TA ElectroForce 3200 series integrating the Optional DMA module. The force data is measured by the optional 22 N load cell, and the displacement data is measured by the integrating displacement sensor (± 7.5 mm, 0.01%FS). Considering the Reviewer’s valuable comment, the modifications made by the authors are as following:

(a) The complete device of the elastic-resistance force test is added into the Supplementary Figure 15.

(b) in the revised manuscript, the detailed information of the test system has been supplemented as: *“In order to compare the elastic-resistance force between the proposed folded and flat magnetic diaphragms, an elastic-resistance force test platform is designed based on the TA ElectroForce system (ElectroForce 3200 series, TA Instruments, BOSE) together with a 22 N load cell (0.2%FS), which can provide the gram-force loads test, as shown in Supplementary Fig. 15. The displacement data is measured by the integrating displacement sensor (range: ± 7.5 mm, 0.03%FS).”* (at the beginning of the Sub-section 4.4 “**Elastic-resistance force test**”)

Reviewer’s Comment 10: *“In Figs. 2c,d and 3i-k, information on the measurement uncertainties is missing. Are these single measurements on a single sample?”*

Authors Response: The authors appreciate the reviewer’s valuable comment. Considering the reviewer’s valuable comment, the modifications made by the authors are as following:

(a) The data in Fig. 2c-d is revised as the average data and the standard deviation from 9 times measurements (3 samples and 3 measurements for each sample).

(b) The data of Fig. 3i-k is the average value from 3 measurements for 1 sample, and its value and standard deviation of 3 measurements is shown in the Supplementary Tables 2-4 as:

Supplementary Table 2. **The average values and standard deviation of the maximum pressure of single diaphragm pump under different magnetic field.**

Magnetic field amplitude (mT)	Pump-in pressure (kPa)		Pump-out pressure (kPa)	
	Average value	Standard deviation	Average value	Standard deviation
0	0.000	0.000	0.000	0.000
5	-0.074	0.005	0.078	0.005
10	-0.135	0.008	0.135	0.004
15	-0.191	0.009	0.195	0.008
20	-0.249	0.008	0.253	0.005
30	-0.351	0.002	0.358	0.007
40	-0.496	0.010	0.501	0.002

Supplementary Table 3. **The average values and standard deviation of the cumulative flow rate of single diaphragm pump under different magnetic field amplitudes and frequencies.**

Magnetic field amplitude (mT)	Flow rate (mL min ⁻¹)							
	1 Hz		2 Hz		5 Hz		10 Hz	
	Average value	Standard deviation	Average value	Standard deviation	Average value	Standard deviation	Average value	Standard deviation
5	4.6	0.1	10.1	0.2	19.4	0.5	41.6	1.0
10	10.9	0.4	19.9	0.7	40.7	0.7	84.4	0.7
15	17.8	0.2	36.7	0.5	72.4	1.0	129.5	1.7
20	24.0	0.6	45.2	0.5	96.8	0.8	178.2	2.6
30	35.1	0.9	63.5	0.8	141.9	0.7		
40	43.7	0.6	86.7	1.2				

Supplementary Table 3. **The average values and standard deviation of the cumulative flow rate of double diaphragm pump under different magnetic field amplitudes and frequencies.**

Magnetic field amplitude (mT)	Flow rate (mL min ⁻¹)							
	1 Hz		2 Hz		5 Hz		10 Hz	
	Average value	Standard deviation	Average value	Standard deviation	Average value	Standard deviation	Average value	Standard deviation
5	16.1	0.2	29.6	0.2	56.4	1.0	103.6	2.2
10	32.1	0.2	58.2	2.1	117.4	3.4	178.1	3.5
15	47.6	1.3	85.0	1.4	180.5	2.2		
20	60.4	0.6	113.6	0.7				
30	75.9	1.0	145.1	1.0				

Reviewer’s Comment 11: “*Information on measurement uncertainties is totally missing in the whole manuscript. Accuracy data in the text of e.g. 178.6 mL/min, 166.67 kPa kg⁻¹, 60066 ml min⁻¹ kg⁻¹ or a “159% higher” deformation are completely implausible.*”

Authors Response: The authors appreciate the Reviewer’s valuable comment. Based on the Reviewer’s valuable comment, in the revised manuscript, the authors added all information regarding the measurement including the uncertainties data as:

(a) In the revised manuscript, the measurement uncertainties of the sensors, including the displacement and force sensors of the TA ElectroForce 3200 system (Subsection 4.4), flow rate sensors (Subsection 4.5), pressure sensors (Subsection 4.5), Gauss meter (Subsection 4.3), is supplemented in the Section 4 “**Methods**” section.

(b) the standard deviations of the measured deformation data between 9 measurements for each type of sample have been added in the Fig. 2c-d and Supplementary Figs. 3-8, and the comparison between the folding and flat circular diaphragms has been revised as: “*For example, the deformation of circular folding diaphragm (5.568±0.118 mm) is 146% larger than that of circular flat diaphragm (2.267±0.114 mm) under same magnetic field*”

(b) the descriptions of the diaphragm pump’s characteristics have been revised as: “*Ignoring the leakage of the check valve, the output pressure and flow rate can be modified by the amplitude of the applied magnetic field, and the flow rate is also related to the frequency of the applied magnetic field, as shown in Fig. 3i-k (the average value and standard deviation data are shown in Fig. 3i-k and summarized in the Supplementary Tables 2-4).* It can be found that under the -10~10 mT harmonic magnetic field with 10Hz frequency, the double diaphragms pump can provide *178.1±3.5 mL min⁻¹ flow rate*, which provides rapid response property. The diaphragm pump merges the lightweight with powerful output properties, and then has large specific pressures (*~167.3±6.7 kPa kg⁻¹, the ratio of maximum pressure to the weight of single diaphragm pump (3.01 g)*) under the 40 mT magnetic field and specific flow rates (*~61394±748 mL min⁻¹ kg⁻¹, the ratio of maximum flow rate to the weight of*

double diaphragm pump (2.94 g) under-15-15 mT harmonic magnetic field with 5 Hz frequency superior to those reported magnetic micro-pumps⁴⁵.” (at the beginning of 1st paragraph in Sub-section 2.3 “**Implementation in diaphragm pump**”, the *Italic* part is the modified part)

The authors would like to express their appreciations again to you for the time and effort you devoted to review our manuscript and to provide us with your constructive suggestions, which we have fully implemented in the revised manuscript.

With sincere appreciation,

Prof. Fan Yang
Huaqiao University, Xiamen, P.R. China
2022.12.01

Authors' Responses to the comments by

Reviewer #3

Manuscript ID: NCOMMS-22-28657

Original Article Title: “Bio-inspired magnetic driving folding diaphragm for biomimetic robot”

Revised Article Title: “Bio-inspired magnetic-driven folded diaphragm for biomimetic robot”

We would like to express our thanks to the reviewer for the very constructive comments made. Please find below a point-by-point reply to each comment. The manuscript has also been revised based on the suggested comments.

General Comment: *“In this manuscript, Yang et al. reported on a magnetically operable soft actuator manufactured in a one-piece mold, which exhibited large, three-dimensional deformation by in response to an external magnetic field with low intensity (~40 mT). This actuator, composed of a mixture of magnetic particles and silicone rubber, was cured in a mold with a folded disk shape and then magnetized in the vertical direction with stretching the disk into a convex form. These processes radially oriented the magnetic moments, so that the actuator could undergo reversible stretching and contraction with turning on and off the vertical magnetic field. The stretching direction could be switched in upward and downward with tuning the magnet direction in upward and downward, respectively, where the folding arrangement enable large deformation with a weak magnetic field. Based on this actuation, various devices, such as gas pump, liquid pump, crowing robot, and swimming robot, were developed.*

My first impression after reading the manuscript is that this is a high-quality work suitable for publication in specialized journal in robotics and engineering. All devises were carefully elaborated to realize good

performances, but the material composition (NdFeB magnetic particle / silicone rubber) and fabrication method (magnetization under elastic deformation) are just an extension of previous works (e.g. ref. 29).

However, I also felt that this work implies one future direction of the field of soft actuators. Although many kinds of magnet-driving soft actuators have been reported, this is the first example of realizing the ‘inside-volume change’ of the actuator; all other examples achieved only the changes in lengths, shapes, and/or angles. Such volume change is the origin of the high performance of the gas/liquid pumps and the swimming robot. In my opinion, the true novelty of this work is not the better mimicking of creatures’ locomotion mechanism, but the realization of ‘inside-volume change’, which would expand the scope of soft actuators.

Therefore, I don’t recommend the current manuscript for publication in the current form, but if it is properly property revised, I think it would become suitable for publication in this journal. Followings are the points to be addressed through the revision.”

Authors Response: The authors appreciate the general comment made by the Reviewer, and will provide response regarding the reviewer’s comments point to point in the following part. The authors would like to express our high gratitude for the general comment made by the Reviewers, especially for “*However, I also felt that this work implies one future direction of the field of soft actuators. Although many kinds of magnet-driving soft actuators have been reported, this is the first example of realizing the ‘inside-volume change’ of the actuator; all other examples achieved only the changes in lengths, shapes, and/or angles. Such volume change is the origin of the high performance of the gas/liquid pumps and the swimming robot. In my opinion, the true novelty of this work is not the better mimicking of creatures’ locomotion mechanism, but the realization of ‘inside-volume change’, which would expand the scope of soft actuators.*” It is exact the essential point of this research paper. In this study, the authors propose a type of one-piece-mold folded diaphragm, consisting of the structure of body segments with radial magnetization property, to achieve large 3-D and bi-direction deformation subjected to simple homogeneous and related low strength magnetic fields, and then

realize large inside-volume changes with high strength. The above properties make this kind of folded diaphragm to be able to be easily customized to conduct different practical applications, such as the pumps and drivers for soft robots.

Considering the Reviewer' valuable comment, the authors made the following modification in the revised manuscript:

- (1) The authors modified the “Abstract” part of the revised manuscript as “Inspired by the locomotion of earthworm, which is conducted through the contraction and stretching between body segments, this study proposes a type of one-piece-mold folded diaphragm, consisting of the structure of body segments with radial magnetization property, to achieve large 3D and bi-directional deformation with inside-volume change capability subjected to the low homogeneous magnetically driving field (40 mT).” (The 2nd sentence in the “Abstract” part of the revised manuscript);
- (2) The authors modified the “Introduction” part of the revised manuscript as “Therefore, in this study, a kind of magnetic-driven folded diaphragm, with one-piece molded and simple manufacturing procedure, and large, 3-D and bi-direction deformation subjected to simple homogeneous and related low strength magnetic fields, is proposed. This kind of folded diaphragm with different radial magnetization properties is able to realize large inside-volume changes with high strength, which is inspired by the locomotion of earthworm generated by the contraction and stretching between body segments.” (The 1st sentence in the last paragraph of the “Introduction” part in the revised manuscript);
- (3) The authors modified the “Conclusion” part of the revised manuscript as “In this study, we propose a type of one-piece-mold folded diaphragm with radial magnetization property, which can achieve large 3D and bi-directional deformation with inside-volume change capability subjected to the low homogeneous magnetically driving field with the easily fabricating method of one-piece mold.” (the 1st sentence of the “Conclusion” part in the revised manuscript).

(4) The authors added “Meanwhile, many kinds of the magnetic-driven soft actuators have been reported, including the origami structure^{22, 30, 36-38}, but most of them can only achieve the changes in shapes³⁰, angles³⁶, and/or lengths³⁸, and cannot realize the relatively large inside-volume changes with high strength, which may limit their potential practical application as soft actuators (drivers), such as the micro-pump.” (the last 3rd sentence in the second paragraph of the “Introduction” part in the revised manuscript).

Here, the authors appreciate again for the general comment made by the Reviewer!

Reviewer’s Comment 1: *“In the introductory part, the difference between the present actuator and conventional magnet-driving soft actuators should be described more clearly.”*

Authors Response: The authors appreciate the Reviewer’s valuable comment. The innovation point and advantages of the proposed actuator is the large, 3-D and bi-direction deformation and inside-volume changes. subjected to simple homogeneous and related low strength magnetic fields. Considering the valuable comment of the reviewer, in the revised manuscript, the authors supplemented the difference between the present actuator and conventional magnet-driving soft actuators in the “**Introduction**” section and added the relative references, as: *“Meanwhile, many kinds of the magnetic-driven soft actuators have been reported, including the origami structure^{22, 30, 36-38}, but most of them can only achieve the changes in shapes³⁰, angles³⁶, and/or lengths³⁸, and cannot realize the relatively large inside-volume changes with high strength, which may limit their potential practical application as soft actuators (drivers), such as the micro-pump.”*

Reviewer’s Comment 2: *“Diaphragm pumps based on other soft actuators have been reported, most of which are based on dielectric elastomers. The characteristics of the present magnet-driving pump compared with conventional soft actuator-based pumps should be clarified.”*

Authors Response: The authors appreciate the reviewer’s valuable comment. Considering the valuable comment from the reviewer, the advantage of the proposed magnetic diaphragm pump is clarified as:

(a) *“Compared with the present magnet-driven pump, which should be driven by the gradient and strongly magnetic field^{41, 42}, the proposed diaphragm pump designed based on the magnetic-driven folded diaphragm pump can realize the large inside-volume change with appreciable loading capability stimulated by the homogeneous and low magnetically driving field, which makes it has considerably protentional application in the area of diaphragm pump. Besides, compared with the conventional soft actuator-based pumps designed based on dielectric elastomers^{43, 44}, the proposed magnetic-driven diaphragm pump can avoid the possible electrical connectors for the application in hydraulic environmental.”* (at the end of the 1st paragraph in Sub-section 2.3 “**Implementation in diaphragm pump**” of the revised manuscript);

(b) *“Furthermore, the co-operation between the magnetic diaphragm and sheets (as check valve) makes the proposed diaphragm pump can be excited by single stimuli without any additional valve device.”* (at the end of the 2nd paragraph in Sub-section 2.3 “**Implementation in diaphragm pump**” of the revised manuscript)

Reviewer’s Comment 3: *“If the design principle for controlling/maximizing the performance of the present diaphragm pump (number of folding repetitions, modulus/thickness of rubber, height of the magnetization mold) is provided, it would be helpful for the readers who want to follow the present strategy.”*

Authors Response: The authors appreciate the Reviewer’s valuable comment. The performance of the present diaphragm pump is related to the deformation properties of the proposed folded diaphragm directly. Considering the reviewer’s valuable comment, in the revised submission the authors supplemented:

(1) the comprehensive test considering different fabrication conditions, including the diaphragm thickness, folding degree, number of segments, hardness of the silicone elastomer, weight ratio of NdFeB particles to silicone elastomer and the magnetization degree, to investigate and analyze the effect of the mechanical parameters on the deformation mechanism of the folding diaphragm, which is related to the performance of the present diaphragm pump;

The test results and analysis of the effect of fabrication conditions on the deformation and elastic-resistance force of the folding diaphragm is provided in the “**Supplementary Discussion 3. The deformation properties of the folded diaphragm**” of the Supplementary File. At the same time, the authors made necessary modification in the revised manuscript as:

(a) adding one paragraph in Sub-section 2.2 “**Mechanical properties**” just before Fig. 2 in the revised manuscript as “Supplementary Discussion 3 summarized the deformation characteristics of the magnetic-driven folded diaphragm subjected to multiple conditions, which include the diaphragm thickness, folded degree, number of segments, hardness of the silicone elastomer, weight ratio of NdFeB particles to silicone elastomer and the magnetization degree. The fabrication conditions are presented in Supplementary Table 1 and the comparison results of different fabrication conditions are shown in Supplementary Fig. 3-8.” (the last paragraph of the Sub-section 2.2 “**Mechanical properties**” in the revised manuscript)

(b) adding two sentences at the end of Sub-section 4.1 “**Material fabrication**” in the revised manuscript to illustrate the testing sample as “In this study, 14 types of prototypes (3 samples for each prototype) with different diaphragm thickness, folded degree, number of segments, hardness of the silicone elastomer, weight ratio of NdFeB particles to silicone elastomer and the magnetization degree, as summarized in Supplementary Table 1, have been fabricated and tested to investigate the deformation and elastic-resistance force properties of the proposed folded diaphragms. Here, it should be noted that each sample (3 samples for each prototype) has been test 3 times, and then total 9 tests for each prototype, as summarized in Fig 2 and Supplementary Fig. 3-8, to verify the”

reproducibility of the deformation properties.” (the last two sentences at the end of Sub-section 4.1 “Material fabrication” in the revised manuscript)

(c) In the revised Supplementary Information, the authors provided the detailed information of the geometric dimensions of structures, which include different folded diaphragms presented in Fig. 2, and pumps as:

(i) Folded diaphragms

Supplementary Figure 12. The size information of four configurations. a Triangle. b Square. c Hexagon. d Circular.

(ii) Pumps

Supplementary Figure 9. The size information of diaphragm pump. a The single diaphragm pump. b The double diaphragm pump

(2) In the revised manuscript, the authors supplemented the comparison between experiments and simulations of the deformation properties of folded and flat diaphragms, as shown in “**Supplementary Discussion 2. Simulations of the folded diaphragm and its applications**” of the Supplementary Information.

Minor Points (i): “Section 3 conclusion: The terms “specific pressure” and “specific flow rate” should be defined.”

Authors Response: The authors appreciate the Reviewer’s valuable comment. In the revised, The terms “specific pressure” and “specific flow rate” have been defined, as: “The diaphragm pump merges the lightweight with powerful output properties, and has large specific pressures ($\sim 167.3 \pm 6.7$ kPa kg⁻¹, the ratio of maximum pressure to the weight of single diaphragm pump (3.01 g)) under the 40 mT magnetic field and specific flow rates ($\sim 61394 \pm 748$ mL min⁻¹ kg⁻¹, the ratio of maximum flow rate to the weight of double diaphragm pump (2.94 g)) under-15-15 mT harmonic magnetic field with 5 Hz frequency superior to those reported magnetic micro-pumps⁴⁵.”(at the middle of the 2nd paragraph in Sub-section 2.3 “**Implementation in diaphragm pump**” of the revised manuscript, and the *Italic* part is the modified part)

Minor Points (ii): “Section 2.1.1: “Fig. 5” should read “Fig. 4”.”

Authors Response: The authors appreciate the Reviewer’s valuable comment. “Fig. 5” has been revised as “Fig. 4” in Sub-section 2.4 “**Bionic earthworm robots**”.

Minor Points (iii): “Section 2.4.2: “Fig. 4” should read “Fig. 5”.”

Authors Response: The authors appreciate the Reviewer’s valuable comment. “Fig. 4” has been revised as “Fig. 5” in Sub-section 2.5 “**Bionic squid swimming robots**”.

Minor Points (iv): “Figure 4: There are two “(b)”.”

Authors Response: The authors appreciate the Reviewer's valuable comment. The number in Fig.4 has been revised.

Minor Points (v): *“Figure 2a, b and Figure 3b–d, f–h : These pictures seem to be edited by some graphic software for easy understanding. However, from scientific and ethics viewpoints, such image editing should be minimized, or ideally should not be done.”*

Authors Response: The authors appreciate the Reviewer's valuable comment. For the Fig. 2a-b in the revised manuscript, the scale during the process of the removing background and equally scale has been supplemented in order to make the reader to compare easily. Besides, the dimensions of four configurations together with its deformation front view under different magnetic field (with scale) has been provided to make the reader to compare easily. Similarly, the scale of the Fig. 3b–d, f–h has been added.

Minor Points (vi): *“Throughout the manuscript: The authors may well be conscious about the house style commonly used in scientific journals, in terms of the use of past/present/future tenses, upper/lower-case letters, space insertion between values and units, etc.”*

Authors Response: The authors appreciate the Reviewer's valuable comment. the authors have double checked and revised the grammar and spelling of the manuscript. Thanks again for the reviewer's comment.

The authors would like to express their appreciations again to you for the time and effort you devoted to review our manuscript and to provide us with your constructive suggestions, which we have fully implemented in the revised manuscript.

With sincere appreciation,

Prof. Fan Yang
Huaqiao University, Xiamen, P.R. China
2022.12.01

REVIEWERS' COMMENTS

Reviewer #1 (Remarks to the Author):

I recommend the revised manuscript for publication in the present form

Reviewer #3 (Remarks to the Author):

I appreciate the efforts made the by the authors to address all scientific points I raised in the previous round of review. The revised version clarifies the difference of the proposed system from conventional soft actuators and pumps made of them, as well as the design principle for desired diaphragm performance. Also, the quantitative information of each device is added, which is indeed necessary for ensuring the scientific solidity of this work.

My remaining concern is about the problem in house style and English. The followings are just examples:

- “type +” and “type –” are expressed in various styles (capitalized/decapitalized, with/without space, long/short hyphen).
- A space between value and unit is often missing, including those embedded in figures.
- page 2, 10th line from the bottom: “the deformation of the folded diaphragm deformation” is not English.
- page 2, 7th line from the bottom: “the present magnet-driven pump” is misleading. The authors want to mean ‘conventional magnet-driven pump’.
- Most of the sentences explaining experimental results are written in the present tense, which is unconventional for scientific writing.

Responses to reviewers' comments

Manuscript ID: NCOMMS-22-28657A

Article Title: “Bio-inspired magnetic-driven folded diaphragm for biomimetic robot”

Dear Editors, Nature Communications

Thank you for reconsidering for publication of our manuscript, with an opportunity to address the reviewers' comments.

We are uploading the revised manuscript, and our point-by-point response to the reviewers' comments.

Best regards,

Prof. Fan Yang
Huaqiao University, Xiamen, P.R. China
2022.12.30

Authors' Responses to the comments by

Reviewer #1

Manuscript ID: NCOMMS-22-28657A

Article Title: "Bio-inspired magnetic-driven folded diaphragm for biomimetic robot"

General Comment: *"I recommend the revised manuscript for publication in the present form"*

Authors Response: The authors would like to express their appreciations again to you for the time and effort you devoted to review our manuscript and to provide us with your constructive suggestions..

With sincere appreciation,

Prof. Fan Yang
Huaqiao University, Xiamen, P.R. China
2022.12.30

Authors' Responses to the comments by

Reviewer #3

Manuscript ID: NCOMMS-22-28657A

Article Title: "Bio-inspired magnetic-driven folded diaphragm for biomimetic robot"

We would like to express our thanks to the reviewer for the very constructive comments made. Please find below a point-by-point reply to each comment. The manuscript has also been revised based on the suggested comments.

General Comment: *"I appreciate the efforts made the by the authors to address all scientific points I raised in the previous round of review. The revised version clarifies the difference of the proposed system from conventional soft actuators and pumps made of them, as well as the design principle for desired diaphragm performance. Also, the quantitative information of each device is added, which is indeed necessary for ensuring the scientific solidity of this work."*

Authors Response: The authors would like to express their appreciations again to you for the time and effort you devoted to review our manuscript and to provide us with your constructive suggestions, which we have fully implemented in the revised manuscript. The authors appreciate the general comment made by the Reviewer, and will provide response regarding the reviewer's comments point to point in the following part.

Reviewer's Comment: *"My remaining concern is about the problem in house style and English. The followings are just examples:*

- “type +” and “type –” are expressed in various styles (capitalized/decapitalized, with/without space, long/short hyphen).
- A space between value and unit is often missing, including those embedded in figures.
- page 2, 10th line from the bottom: “the deformation of the folded diaphragm deformation” is not English.
- page 2, 7th line from the bottom: “the present magnet-driven pump” is misleading. The authors want to mean ‘conventional magnet-driven pump’.
- Most of the sentences explaining experimental results are written in the present tense, which is unconventional for scientific writing.”

Authors Response: The authors appreciate the general comment made by the Reviewer. The authors have done our best to revise the whole paper to correct the typos, make the house style and English of whole paper satisfied the requirement of the Journal, especially in terms of the use of past/present/future tenses, upper/lower-case letters, space insertion between values and units, etc. Following, it is the authors’ response point-to-point regarding the reviewer’s comments.

Points (i): “type +” and “type –” are expressed in various styles (capitalized/decapitalized, with/without space, long/short hyphen).”

Authors Response: The authors appreciate the reviewer’s valuable comment. The formats of the “Type+” and “Type–” have been unified in the whole manuscript, including those embedded in figures.

Points (ii): “A space between value and unit is often missing, including those embedded in figures.”

Authors Response: The authors appreciate the reviewer’s valuable comment. the authors double checked the space between value and unit, including those embedded in figures, and made all corrections.

Points (iii): *“page 2, 10th line from the bottom: “the deformation of the folded diaphragm deformation” is not English.”*

Authors Response: The authors appreciate the reviewer’s valuable comment. In the revised manuscript, the sentence has been revised as: “Under vertical downward magnetic field conditions, the deformation of the folded diaphragm compresses the cavity volume, and then pumps the flow (gas/water) out from the pump cavity, as shown in Fig. 3d, h.” Furthermore, the authors checked the whole paper and corrected typos.

Points (iv): *“page 2, 7th line from the bottom: “the present magnet-driven pump” is misleading. The authors want to mean ‘conventional magnet-driven pump’.”*

Authors Response: The authors appreciate the reviewer’s valuable comment. In the revised manuscript, the sentence has been revised as: “Compared with the conventional magnetic-driven pump.....”

Points (v): *“Most of the sentences explaining experimental results are written in the present tense, which is unconventional for scientific writing.”*

Authors Response: The authors appreciate the reviewer’s valuable comment. the authors modified the whole manuscript with proper tense, especially Section “Methods”, in which the past tense has been applied.

The authors would like to express their appreciations again to you for the time and effort you devoted to review our manuscript and to provide us with your constructive suggestions, which we have fully implemented in the revised manuscript.

With sincere appreciation,

Prof. Fan Yang
Huaqiao University, Xiamen, P.R. China
2022.12.30